# Cellular stress promotes NOD1/2-dependent inflammation via the endogenous metabolite sphingosine-1-phosphate

Gang Pei[1,*,†] iD, Joanna Zyla[1,2] iD, Lichun He[3,4], Pedro Moura-Alves[1,5] iD, Heidrun Steinle[6], Philippe Saikali[1], Laura Lozza[1], Natalie Nieuwenhuizen[1], January Weiner[1], Hans-Joachim Mollenkopf[7], Kornelia Ellwanger[6], Christine Arnold[6] iD, Mojie Duan[3,4], Yulia Dagil[8], Mikhail Pashenkov[8], Ivo Gomperts Boneca[9,10,11], Thomas A Kufer[6] iD, Anca Dorhoi[12,13,**] iD & Stefan HE Kaufmann[1,14,***] iD

## Abstract

Cellular stress has been associated with inflammation, yet precise underlying mechanisms remain elusive. In this study, various unrelated stress inducers were employed to screen for sensors linking altered cellular homeostasis and inflammation. We identified the intracellular pattern recognition receptors NOD1/2, which sense bacterial peptidoglycans, as general stress sensors detecting perturbations of cellular homeostasis. NOD1/2 activation upon such perturbations required generation of the endogenous metabolite sphingosine-1-phosphate (S1P). Unlike peptidoglycan sensing via the leucine-rich repeats domain, cytosolic S1P directly bound to the nucleotide binding domains of NOD1/2, triggering NF-κB activation and inflammatory responses. In sum, we unveiled a hitherto unknown role of NOD1/2 in surveillance of cellular homeostasis through sensing of the cytosolic metabolite S1P. We propose S1P, an endogenous metabolite, as a novel NOD1/2 activator and NOD1/2 as molecular hubs integrating bacterial and metabolic cues.

**Keywords** NOD-like receptors; inflammation; sphingolipid metabolism; cellular homeostasis; NOD1/2

**Subject Categories** Immunology; Metabolism
**The EMBO Journal (2021) 40: e106272**
See also: Y Lu & D Neculai (July 2021)

## Introduction

Nucleotide-binding oligomerization domain-containing protein 1/2 (NOD1/2) are intracellular pattern recognition receptors that activate innate immune responses by sensing bacterial peptidoglycans (Caruso *et al*, 2014; Philpott *et al*, 2014). In addition to well-established roles in bacterial sensing, both NOD1 and NOD2 modulate immune responses to a broad range of peptidoglycan-free microbes, such as respiratory syncytial virus (Sabbah *et al*, 2009), vesicular stomatitis virus (Sabbah *et al*, 2009), influenza A virus (Lupfer *et al*, 2014), human cytomegalovirus (Fan *et al*, 2016), hepatitis C virus (Vegna *et al*, 2016), *Plasmodium berghei* (Finney *et al*, 2009), *Plasmodium falciparum* (Corbett *et al*, 2015), and *Trypanosoma cruzi* (Silva *et al*, 2010). Disruption of actin dynamics by chemicals (Legrand-Poels *et al*, 2007; Bielig *et al*, 2014) and manipulation of

1 Department of Immunology, Max Planck Institute for Infection Biology, Berlin, Germany
2 Department of Data Science and Engineering, Silesian University of Technology, Gliwice, Poland
3 State Key Laboratory of Magnetic Resonance and Atomic Molecular Physics, Key Laboratory of Magnetic Resonance in Biological Systems, National Center for Magnetic Resonance in Wuhan, Wuhan Institute of Physics and Mathematics, Innovation Academy for Precision Measurement Science and Technology, Chinese Academy of Sciences, Wuhan, China
4 University of Chinese Academy of Sciences, Beijing, China
5 Nuffield Department of Medicine, Ludwig Institute for Cancer Research, University of Oxford, Oxford, UK
6 Department of Immunology, Institute of Nutritional Medicine, University of Hohenheim, Stuttgart, Germany
7 Microarray Core Facility, Max Planck Institute for Infection Biology, Berlin, Germany
8 Institute of Immunology of the Federal Medical-Biological Agency of Russia, Moscow, Russia
9 Institut Pasteur, Department of Microbiology, Biology and Genetics of the Bacterial Cell Wall, Paris, France
10 CNRS UMR2001, Integrative and Molecular Microbiology, Paris, France
11 INSERM, Équipe AVENIR, Paris, France
12 Institute of Immunology, Friedrich-Loeffler-Institut, Greifswald-Insel Riems, Germany
13 Faculty of Mathematics and Natural Sciences, University of Greifswald, Greifswald, Germany
14 Hagler Institute for Advanced Study at Texas A&M University, College Station, TX, USA
    *Corresponding author. Tel: +49 383517 4978; E-mail: gang.pei@fli.de
    **Corresponding author. Tel: +49 383517 1624; E-mail: anca.dorhoi@fli.de
    ***Corresponding author. Tel: +49 3028460 500/502; E-mail: kaufmann@mpiib-berlin.mpg.de
    †Present address: Institute of Immunology, Friedrich-Loeffler-Institut, Greifswald-Insel Riems, Germany

Rho GTPases by pathogens (Keestra *et al*, 2013) also induce NOD1/2-dependent inflammatory responses. Additionally, chemical- or infection-induced endoplasmic reticulum (ER) stress triggers NOD1/2-mediated inflammation, as well (Keestra-Gounder *et al*, 2016). Whether NOD1/2 detect other types of cellular stress and what enables these receptors to sense such diverse stimuli remain unknown.

Sphingolipids are a structurally diverse class of lipids characterized by a long-chain amino backbone. They are essential structural components of plasma membrane and important mediators involved in diverse biological pathways (Hannun & Obeid, 2008; Bartke & Hannun, 2009). Sphingosine, ceramide, ceramide-1-phosphate (C1P), sphingosine-1-phosphate (S1P), and other sphingolipid derivatives regulate apoptosis (Birbes *et al*, 2001; Ganesan *et al*, 2010), cell proliferation (Zhang *et al*, 1990), inflammation (Maceyka & Spiegel, 2014; Hannun & Obeid, 2018), and autophagy (Young & Wang, 2018). Extracellular S1P binds to the cell surface receptors $S1PR_{1-5}$ and regulates immune cell trafficking (Spiegel & Milstien, 2003, 2011; Rosen & Goetzl, 2005; Maceyka & Spiegel, 2014). S1P also functions as an intracellular signaling molecule and interferes with histone acetylation (Hait *et al*, 2009), calcium mobilization from the ER (Ghosh *et al*, 1990, 1994; Mattie *et al*, 1994), and TNFα-induced NF-κB activation (Alvarez *et al*, 2010), which indicates pleiotropic targets of S1P within the cell.

Using knockouts and inducible NOD1/2 expression systems, we report that diverse stress stimuli induce NOD1/2-RIP2-dependent inflammatory responses. Gene expression analysis revealed that key enzymes involved in the sphingolipid pathway were induced by various stressors. Further, lipidomic profiling demonstrated that cytosolic production of S1P, among other lipid metabolites, was elevated in such conditions. This lipid mediator was essential for NOD1/2 activation. The cytosolic delivery of S1P triggered NOD1/2-RIP2-dependent NF-κB activation and pro-inflammatory responses. Mechanistic studies elucidated that S1P specifically and directly bound to the nucleotide binding domains (NBDs) of NOD1/2, independent of the leucine-rich repeats (LRR) domains. Altogether, we established that sphingolipid metabolism governs inflammation triggered by cellular stress and identified a central role for S1P in triggering NOD1/2-dependent inflammation upon stress induction.

# Results

## Various kinds of stress induce NOD1/2-dependent inflammatory responses

We postulated that NOD1/2 function as sensors of perturbation of cellular homeostasis. To address this hypothesis, we generated HeLa doxycycline-inducible NOD1 or NOD2 cells specific for NOD1 or NOD2 agonists, respectively (Appendix Fig S1A–C). We stimulated these cells with various chemicals that interfere with cellular homeostasis through distinct pathways, including perturbation of microtubule (paclitaxel, vinblastine, nocodazole, colchicine) or actin (jasplakinolide, cytochalasin D) dynamics; induction of Golgi (nigericin, monensin, brefeldin A), mitochondrial (actinonin, CCCP, lovastatin), or ER stress (tunicamycin, thapsigargin); and protein translation inhibition (cycloheximide, anisomycin) and DNA damage (etoposide). In the rest of this study, we chose 1 or 2

representative chemicals from each class for the stimulation experiments. We observed that induction of *IL6* and *IL8* expression and production correlated with NOD1/2 expression, without considerable differences of cell death in association with NOD1 or NOD2 expression (Fig 1A–D and Appendix Fig S1D–G). Transcription profiling revealed that expression of genes involved in NF-κB activation and inflammatory responses was commonly induced by these stimulations (Fig 1E and F). To exclude effects of doxycycline itself, we generated HeLa doxycycline-inducible GFP cells and stimulated them with the same chemicals. Compared to HeLa NOD1 or NOD2 cells, IL6 production stimulated by these compounds was markedly diminished and doxycycline-dependent IL6 production was not observed (Fig EV1A). A recent study showed that ER stress triggers endocytosis of the trace peptidoglycan contaminants in the serum, thereby inducing pro-inflammatory signaling (Molinaro *et al*, 2019). By using serum-free medium Opti-MEM (Fig EV1B and C) and the inhibitor of endocytosis-Dynasore (Fig EV1D and E), we revealed that the high IL6 production in response to most of the stimulations above was not blocked by serum depletion or endocytosis inhibition, thus excluding the possibility of peptidoglycan contaminants in the serum. We conclude that perturbation of cellular homeostasis induces specific NOD1/2-dependent pro-inflammatory signaling.

## Cellular stress triggers NOD1/2-RIP2-dependent NF-κB activation and MAPK activation

To decipher mechanisms underlying NOD1/2-dependent signaling, NF-κB reporter cells were established on the background of HEK293T wild type (WT) and NOD1/2 double knockout (dKO). NF-κB luciferase assays confirmed that perturbation of cellular homeostasis induced NF-κB activation, while NF-κB activation and *IL8* expression were significantly impaired by *NOD1/2* dKO (Fig 2A and B). Consistently, *Cxcl2* expression and production of CXCL2, CCL2, and IL16 in bone marrow-derived macrophages (BMDMs) upon perturbation of cellular homeostasis were compromised in both *Nod1/2* dKO and *Rip2* KO cells (Figs 2C and D, and EV2A and B). Accordingly, NF-κB activation and MAPK activation upon ER stress in BMDMs were also impaired in both *Nod1/2* dKO and *Rip2* KO BMDMs (Fig EV2C and D). We conclude that disruption of cellular homeostasis triggers NF-κB activation and MAPK activation in a NOD1/2-RIP2-dependent manner.

## S1P generation is required for inflammation induced by perturbation of cellular homeostasis or by intracellular Shigella infection

To uncover changes in sphingolipid metabolism upon perturbation of cellular homeostasis, primary human dermal fibroblasts (HDFs) were employed for two reasons: (i) Sphingolipid metabolism in cancer cells is dysregulated (Ryland *et al*, 2011; Ogretmen, 2018) and (ii) NOD1 and NOD2 are strongly expressed in different sources of human primary fibroblasts (Uehara & Takada, 2007; Hirao *et al*, 2009; Jeon *et al*, 2012). Numerous stimuli induce generation of bioactive sphingolipids, including ceramide, C1P, sphingosine, and S1P (Hannun & Obeid, 2008, 2018; Bartke & Hannun, 2009). Indeed, in HDFs the expression of *ACER1/2, SMPD3, SPHK1* and other key enzymes involved in sphingolipid metabolism (Fig EV3A) were

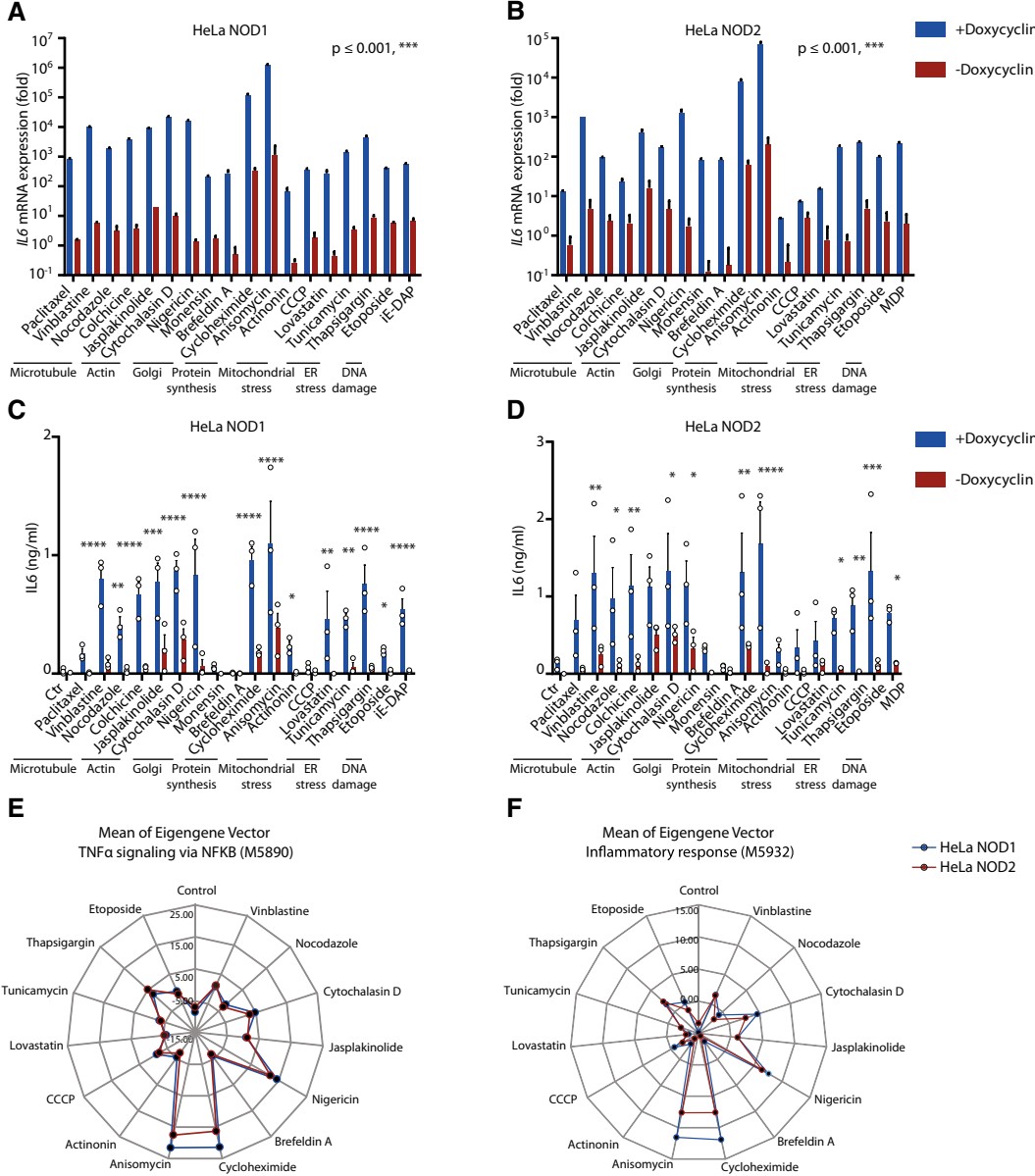

**Figure 1. NOD1/2 sense perturbation of cellular homeostasis.**

A, B qRT–PCR analysis of *IL6* expression upon indicated stimulations in the presence or absence of NOD1 (A) or NOD2 (B). HeLa inducible NOD1 cells (A) or NOD2 cells (B) were induced in the absence or presence of doxycycline overnight and afterward stimulated with various stimuli for 4 h.

C, D ELISA analysis of IL6 production in supernatants of HeLa NOD1 cells (C) or NOD2 cells (D) upon different stimulations. HeLa NOD1 cells (C) or NOD2 cells (D) were induced in the absence or presence of doxycycline overnight and stimulated with various stimuli for 20 h, and afterward, supernatants were collected for ELISA.

E, F Sample mean value of eigengene analysis of TNFα signaling via NF-κB pathway (E) and inflammatory response pathway (F) of HeLa NOD1 or NOD2 cells upon indicated stimulations.

Data information: (A, B) Means ± SD of three technical replicates from one representative experiment out of four independent experiments. *P* values were calculated with ANOVA test of linear mixed model for fix effect of main factors. (C, D) Means ± SEM of three independent experiments. Each dot represents one independent experiment. *P* values were calculated using two-way ANOVA. *$P \leq 0.05$, **$P \leq 0.01$, ***$P \leq 0.001$, and ****$P \leq 0.0001$.

induced by various stressors (Fig 3A). Consistently, lipidomic profiling unveiled elevated production of cellular sphingosine (d18:1) and S1P (d16:1, d17:1, d18:1) in HDFs upon various stressors (Fig 3B), whereas the abundance of other lipid classes remained unaffected (Fig EV3B). S1P ELISA further confirmed that the abundance of S1P upon stimulation with various stressors was increased up to 3 μM

(Fig 3C). To interrogate whether sphingolipid metabolism contributes to stress-induced inflammation, THP-1 cells, BMDMs, and human CD14[+] monocytes were treated with inhibitors targeting distinct steps of the sphingolipid pathway and *IL6* expression and production were evaluated upon tunicamycin-induced ER stress (Fig EV3C–G). In all types of cells, *IL6* expression and production

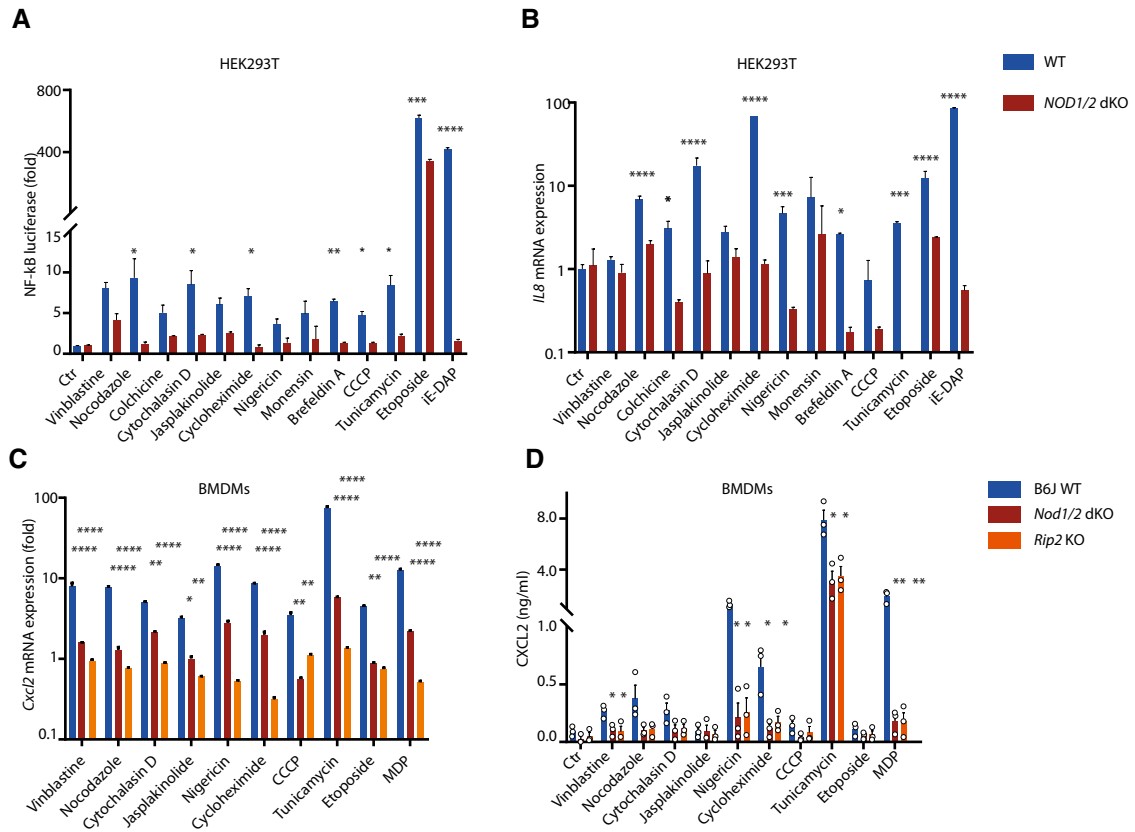

**Figure 2. NOD1/2 signaling is required for NF-κB activation and inflammatory response upon perturbation of cellular homeostasis.**

A  NF-κB luciferase assay in HEK293T wild-type (WT) and *NOD1/2* knockout (KO) NF-κB reporter cells upon various stimulations.
B  qRT–PCR analysis of *IL8* expression upon indicated stimulations in HEK293T WT and *NOD1/2* KO cells.
C  qRT–PCR analysis of *Cxcl2* expression upon indicated stimulations in WT, *Nod1/2* KO, and *Rip2* KO bone marrow-derived macrophages (BMDMs).
D  Multiplex analysis of CXCL2 production in supernatants of WT, *Nod1/2,* and *Rip2* KO BMDMs upon indicated stimulations for 20 h.

Data information: (A-C) Means ± SD of three technical replicates from one representative experiment out of three independent experiments. (D) Means ± SEM of three independent experiments. Each dot represents one independent experiment. *P* values were calculated using two-way ANOVA. *$P \leq 0.05$, **$P \leq 0.01$, ***$P \leq 0.001$, and ****$P \leq 0.0001$.

were abolished by the ceramidase inhibitor ceranib-2 and by the sphingosine kinase inhibitor SKI-II (Fig EV3C–G), suggesting that S1P production is essential for ER stress-induced IL6 production. Moreover, *IL6* expression and production in response to various stressors in HDFs and HeLa inducible NOD1 or NOD2 cells were markedly diminished by inhibition of sphingosine kinases (Fig 3D–I). In humans and mice, two genes (*Sphk1* and *Sphk2*) encode the sphingosine kinase (Adams *et al,* 2016). Due to relatively low IL6 production by BMDMs upon other stress stimuli as observed before (Keestra-Gounder *et al,* 2016), we employed *Sphk1* and *Sphk2* KO BMDMs and validated whether *Sphk1* and *Sphk2* contributed to stress-induced CXCL2 expression and production. Consistent with a more pronounced decrease of S1P in *Sphk2* KO cells (Canlas *et al,* 2015; Zhang *et al,* 2015), *Sphk2* KO caused more profound reduction in CXCL2 expression and production (Fig 3H and I). MAPK activation induced by ER stress was also reduced by *Sphk2* KO (Fig EV3H and I). To investigate the role of S1P production in bacterial-induced activation of NOD1/2, we infected HeLa cells with the Gram-negative enteropathogenic bacterium *Shigella flexneri*. SPHK1/2 double knockdown (dKD) resulted in lower IL8 production upon Shigella

infection, suggesting the involvement of S1P in Shigella-induced NF-κB activation (Figs 3J and EV3J). Together, we conclude that S1P generation is required for inflammation induced by perturbation of cellular homeostasis and by intracellular Shigella infection.

**S1P is produced through the hydrolysis pathway upon ER stress**

S1P is mainly generated through the *de novo* synthesis or by the hydrolysis pathway (Spiegel & Milstien, 2003; Hannun & Obeid, 2018). Serine palmitoyltransferase (SPTLC1) and sphingomyelinase (SMPD) are the first enzymes involved in the *de novo* synthesis or the hydrolysis pathway, respectively. To interrogate through which pathway S1P is produced upon stress induction, the two enzymes essential for *de novo* synthesis or the hydrolysis pathway were blocked by specific chemical inhibitors (Fig EV3A). *IL6* expression and production in THP-1 cells upon ER stress were abrogated by acidic sphingomyelinase inhibitor imipramine, but not by inhibition of serine palmitoyltransferase (L-cycloserine) or neutral sphingomyelinase (GW4869), uncovering that S1P is mainly produced via the hydrolysis pathway upon ER stress (Appendix Fig S2).

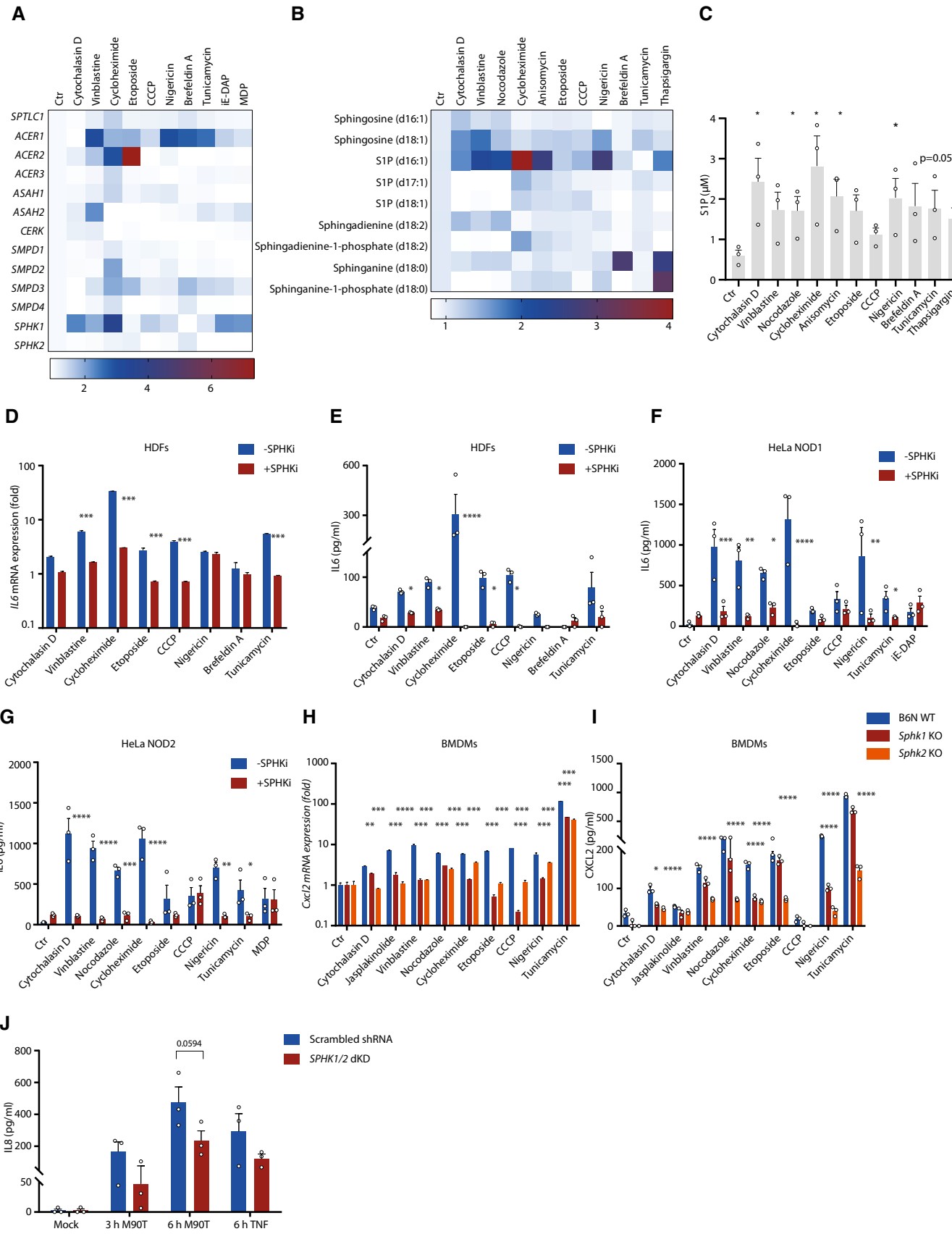

Figure 3.

**Figure 3. S1P generation is essential for pro-inflammatory responses induced by stressors and by Shigella infection.**

A   Fluidigm analysis of expression of genes involved in sphingolipid metabolism in human dermal fibroblasts (HDFs) upon indicated stimulations for 4 h. Heatmap shows means of 4 independent experiments. The color scale represents the relative gene expression compared to control from lower (blue) to higher levels (red).

B   Lipidomic profiling of sphingolipid metabolites in HDFs upon indicated stimulations for 2 h. Fold changes of each metabolite were calculated against corresponding controls. Heatmap shows means of three independent experiments. The color scale represents the relative levels of various lipids from lower (blue) to higher abundance (red).

C   ELISA analysis of cytosolic S1P production in HDFs upon indicated stimulation for 4 h.

D   qRT–PCR analysis of *IL6* expression in HDFs upon indicated stimulations for 4 h in the absence or presence of the inhibitor of sphingosine kinases-SKI-II (SPHKi, 50 μM).

E–G   ELISA analysis of IL6 in supernatants of HDFs (E), HeLa NOD1 cells (F), or HeLa NOD2 cells (G) upon indicated stimulations for 20 h in the absence or presence of the inhibitor of sphingosine kinases.

H   qRT–PCR analysis of *Cxcl2* expression in WT, *Sphk1* KO, or *Sphk2* KO BMDMs upon indicated stimulations for 4 h.

I   Multiplex analysis of CXCL2 production in supernatants of WT, *Sphk1*, and *Sphk2* KO BMDMs upon indicated stimulations for 20 h.

J   ELISA analysis of IL8 production upon Shigella infection. HeLa scrambled control cells and *SPHK1/2* double KD cells were infected with *Shigella flexneri* M90T for 3 and 6 h or stimulated with TNFα for 6 h.

Data information: (C, E-G, I, J) Means ± SEM of three independent experiments. Each dot represents one independent experiment. (D, H) Means ± SD of three technical replicates from one representative experiment out of three independent experiments. *P* values were calculated using one-way or two-way ANOVA. *$P \leq 0.05$, **$P \leq 0.01$, ***$P \leq 0.001$, and ****$P \leq 0.0001$.

## Cytosolic S1P induces NOD1/2-RIP2-mediated NF-κB activation and inflammation

To investigate whether S1P can activate NOD1/2, cells were stimulated with extracellular and cytosolic S1P. Extracellular S1P stimulation did not induce marked IL6 or IL8 production (Fig EV4A–D). In contrast, cytosolic delivery of S1P with digitonin induced abundant IL6 or IL8 production in a NOD1/2-dependent manner (Fig 4A and B). Compared to the known NOD1/2 activators iE-DAP or MDP, cytosolic S1P induced higher IL6 production at the same concentration (Fig EV4E and F). Similar to iE-DAP or MDP, S1P stimulation did not induce NOD1 or NOD2 degradation (Fig EV4G).

The biological activities of extracellular S1P are mediated by S1PRs, a group of G protein-coupled receptors which regulate immune cell trafficking, cytoskeleton reorganization, and cell proliferation upon S1P binding (Rosen & Goetzl, 2005). To address the involvement of S1PRs in S1P-induced NOD1/2 activation, specific inhibitors against S1PR$_1$ (W146), S1PR$_2$ (JTE013), S1PR$_1$ and S1PR$_3$ (VPC23019), and S1PR$_4$ (CYM50358), as well as the general S1PR inhibitor FTY720 or blocking GPCR signaling by pertussis toxin, were employed. All inhibitors failed to block IL6 production by cytosolic S1P (Appendix Fig S3A and B). Cytosolic S1P is irreversibly degraded by S1P lyase, yielding ethanolamine phosphate and the long-chain aldehyde trans-2-hexadecenal (Bourquin *et al*, 2010). Inhibiting S1P lyase by its specific inhibitor, THI resulted in higher IL6 production by S1P, but not by iE-DAP or MDP, further confirming the roles of cytosolic S1P in NOD1/2 activation (Appendix Fig S3C and D). Notably, other metabolites involved in sphingolipid metabolism, such as C2 ceramide (10 μM), C16 ceramide (10 μM), and sphingosine (10 μM), did not induce IL6 production, indicating specificity of S1P as lipid species able to trigger innate activation (Appendix Fig S3E and F). Thus, cytosolic S1P specifically induces NOD1/2-dependent inflammatory response in a S1PR independent manner.

To interrogate the downstream signaling of NOD1/2 activation by cytosolic S1P, NOD1 and NOD2 were immunoprecipitated and Western blotting was employed for detecting RIP2. Cytosolic S1P stimulated the interaction between NOD1/2 and RIP2, indicating activation of NOD1/2 signaling via RIP2 (Fig 4C). Blocking the interaction between NOD1/2 and RIP2 by the pharmacological inhibitor-GSK583 abolished IL6 production by cytosolic S1P (Appendix Fig S3A and B). Consistently, cytosolic S1P triggered MAPK activation and NF-κB activation (Figs 4D and Appendix Fig S3G). Furthermore, cytosolic S1P induced NOD1/2-dependent NF-κB activation and *IL8* expression in HEK293T cells (Fig 4E and F). Finally, *Il6* expression and production of IL6, CXCL2, CCL5, and IL16 induced by cytosolic S1P were markedly impaired in *Nod1/2* KO and *Rip2* KO BMDMs (Fig 4G–K). We conclude that cytosolic S1P induces NOD1/2-RIP2-mediated NF-κB activation and inflammation.

## S1P directly binds to NOD1/2 via NBDs

We hypothesized that S1P may directly interact with NOD1/2. To this end, immunoprecipitations were performed with different lipid-coated beads. NOD1 and NOD2 were specifically immunoprecipitated with S1P-coated beads, demonstrating interactions between S1P and NOD1/2 (Fig 5A and B). Immunoprecipitation with THP-1 cells further confirmed the interaction between S1P and endogenous NOD1 (Fig 5C). Direct stochastic optical reconstruction microscopy (dSTORM) demonstrated the oligomerization of NOD1/2 induced by S1P (Fig EV5A). Dual-color dSTORM demonstrated that cytosolic S1P colocalized with NOD1/2 (Fig 5D). Single particle tracking in live cells further demonstrated that S1P colocalized with NOD1/2 and moved together with NOD1/2, confirming its interaction with NOD1/2 inside living cells (Fig 5E and F).

To delineate potential S1P binding sites, we performed *in silico* docking on the rabbit NOD2 crystal structure (PDB code: 5IRN) with the S1P molecule using AutoDock (Morris *et al*, 2009). The docking result revealed the accommodation of S1P in the pocket of NOD2, corresponding to the NACHT/NBD. Residues in the NBD, such as K285, S286, R314, D359, T404, E580, and H583, were predicted to interact with S1P (Fig 6A). Furthermore, molecular dynamics simulation of 10 ns was carried out using the docked complex and showed almost no movement of the S1P, validating the docking result. To determine whether NOD1/2 directly interacts with S1P, NOD1/2 proteins were purified (Fig 6B) and microscale thermophoresis (MST) assays were performed with purified proteins

(Fig 6C–E). MST experiments revealed direct interactions of S1P with NOD1 or NOD2, which was not observed with GFP or the NOD-like receptor NLRP3 (Fig 6C). The purified NOD1 or NOD2 was specific for their bacterial ligands iE-DAP and MDP, respectively (Fig EV5B and C). The $K_d$ value of S1P interacting with NOD1 or NOD2 was around 2 or 5 µM, respectively. An interaction of C16 ceramide with NOD1 or NOD2 was not detected (Fig EV5B and C).

These findings demonstrate the specificity of S1P interaction with NOD1 or NOD2. Intriguingly, the $K_d$ value of S1P with NOD1/2 was lower than the values of the canonical NOD1/2 agonists iE-DAP or MDP in our system (Fig 6D and E). To substantiate our docking results regarding binding of S1P to the NACHT/NBDs of NOD1/2, HEK293T NOD1/2 KO cells were transfected with different truncated NOD1/2 mutants and the IL8 release was evaluated upon S1P

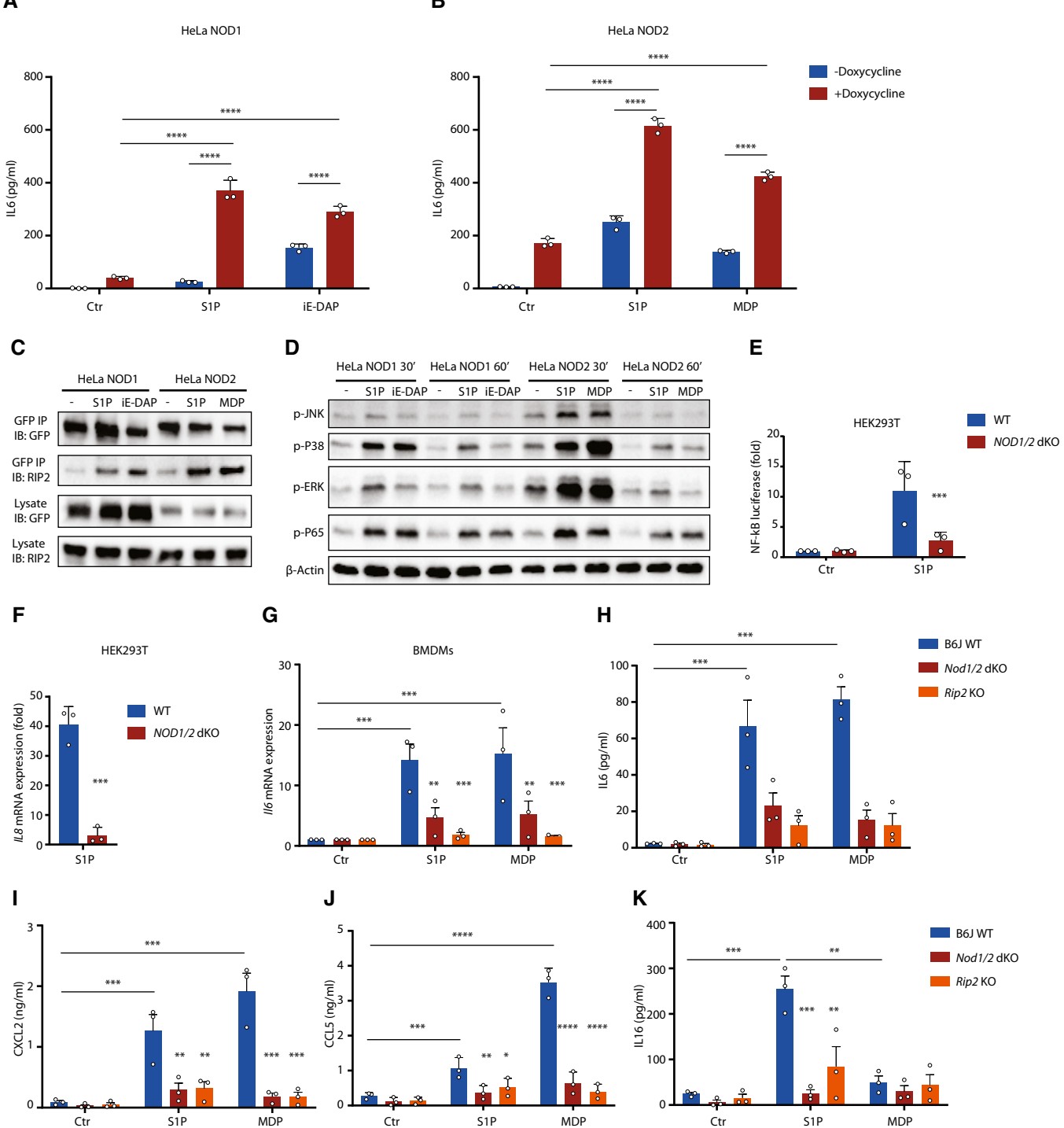

**Figure 4.**

**Figure 4.  Cytosolic S1P-induced NF-κB activation and inflammation is mediated by NOD1/2.**

A, B  ELISA analysis of IL6 in HeLa NOD1 cells (A) or HeLa NOD2 cells (B) upon indicated stimulations for 20 h. For cytosolic delivery of S1P, iE-DAP, or MDP, cells were treated together with digitonin (5 μg/ml).

C  GFP immunoprecipitation (IP) of NOD1/2 upon S1P, iE-DAP, or MDP stimulation. S1P (20 μM), iE-DAP (20 μM), or MDP (20 μM) was delivered into the cytosol of HeLa NOD1 or HeLa NOD2 cells with digitonin (5 μg/ml) for 1 h. Afterward, cell lysates were collected for IP.

D  Western blot analysis of MAPK and NF-κB activation upon S1P stimulation. HeLa NOD1 or NOD2 cells were treated with cytosolic S1P (20 μM), iE-DAP (20 μM), or MDP (20 μM) for indicated times.

E  NF-κB luciferase assay in HEK293T WT and *NOD1/2* KO NF-κB reporter cells upon cytosolic S1P stimulation.

F  qRT–PCR analysis of *IL8* expression upon cytosolic S1P stimulation in HEK293T WT and *NOD1/2* KO cells.

G  qRT–PCR analysis of *Il6* expression in WT, *Nod1/2*, and *Rip2* KO BMDMs upon S1P (10 μM) or MDP (10 μM) stimulations together with digitonin (2.5 μg/ml) for 4 h.

H–K  Multiplex analysis of IL6 (H), CXCL2 (I), CCL5 (J), and IL16 (K) production in supernatants of WT, *Nod1/2*, and *Rip2* KO BMDMs upon S1P (10 μM) or MDP (10 μM) stimulations together with digitonin (2.5 μg/ml) for 20 h.

Data information: (A, B, E-G) Means ± SD of three independent experiments. (H-K) Means ± SEM of three independent experiments. Each dot represents one independent experiment. *P* values were calculated using two-way ANOVA (A, B, E, G-K) or Student's *t*-test (F). *$P \leq 0.05$, **$P \leq 0.01$, ***$P \leq 0.001$, and ****$P \leq 0.0001$. Source data are available online for this figure.

stimulation. IL8 production was induced by S1P in cells expressing NOD1/2 WT. IL8 production in cells expressing LRR-deficient NOD1/2 (NOD1/2ΔLRR) was also elevated upon S1P stimulation, but not upon iE-DAP/MDP stimulation, suggesting that the LRR domains of NOD1/2 are not essential for the interaction with S1P (Fig 6F). Consistently, MST assays demonstrated that S1P still interacted with NOD1ΔLRR and NOD1ΔCARD, confirming that the NBD was responsible for the interaction with S1P (Fig EV5D). To address whether S1P competes with bacterial agonists for the binding to NOD1/2, we performed MST assays using purified NOD1/2 to measure their binding affinities with iE-DAP or MDP in the presence or absence of S1P. The binding affinities of NOD1/2 for iE-DAP or MDP were not affected by S1P, further confirming that binding sites of S1P and bacterial agonists are sterically independent (Fig 6G and H). This further strengthens our observations that S1P binds to the NBD, but not the LRR domains of NOD1/2. To investigate whether co-delivery of S1P with iE-DAP or MDP had synergistic or additive effects on NOD1/2 activation, HeLa NOD1 or NOD2 cells were co-stimulated with S1P and various concentration of iE-DAP or MDP. Indeed, co-stimulation of S1P with iE-DAP or MDP induced significantly higher IL6 production compared to single stimulation alone, indicating independent activating mechanisms of NOD1/2 upon S1P and iE-DAP or MDP stimulation (Fig EV5E and F). Crohn's disease-associated mutation NOD2 1007fs results in a partial deletion in the LRR region and hence defective detection of MDP (Hugot *et al*, 2001; Ogura *et al*, 2001). Based on our data, S1P should still be able to induce activation of NOD2 1007fs. Indeed, IL8 production in HEK293T cells expressing NOD2 1007fs was increased upon S1P stimulation, demonstrating the activation of NOD2 1007fs by S1P (Fig EV5G).

To identify the specific amino acids involved in the interaction, various conserved amino acids in the NBD region of human NOD1 were mutated (Fig EV5H). Cells expressing these mutants were stimulated by S1P or iE-DAP to identify mutations that specifically abolish S1P activation. ELISA analysis showed that K328A or H517A could be activated by iE-DAP but not by S1P. Intriguingly, iE-DAP induces higher IL8 production in cells expressing K328A or H517A than WT, further confirming the different activation mechanisms by S1P and iE-DAP stimulation (Fig EV5H). Consistent with the *in silico* docking, this result suggests that H517 in the winged-helix (WH) domain of human NOD1 is critical for the interaction with S1P (Fig EV5H). Assuming that the conserved H517 residue

interacts with the β-phosphate of ADP (Maekawa *et al*, 2016), we interrogated whether S1P and ADP compete with each other for binding to NOD1/2. To this end, the binding affinities of ADP to NOD1/2 were evaluated in the presence or absence of S1P. In line with a previous report (Askari *et al*, 2012), the $K_d$ values of ADP alone with NOD1 or NOD2 were around 100 or 150 nM, respectively. However, the $K_d$ values of ADP with NOD1 or NOD2 were increased to 1.5 μM by S1P (Fig 7A and B). Similarly, the binding affinities of S1P for NOD1/2 were also compromised by the presence of ADP (Fig 7C and D). Further, MST assays demonstrated that the $K_d$ value of NOD1 H517A with S1P was increased to around 15 μM, suggesting impaired binding of S1P to NOD1 H517A. The ADP binding was also compromised by H517A mutation (Figs 7E and F, and EV5I). Thus, S1P directly binds to NOD1/2 via interaction with H517 and subsequently impairs ADP binding. Hence, S1P likely replaces ADP from the NBD to trigger NOD1/2 activation.

## Discussion

In this study, we demonstrate that perturbation of cellular homeostasis induces production of the cellular metabolite S1P, which binds to NOD1/2 and triggers NOD1/2-dependent inflammation. Hence, NOD1/2 not only sense microbial ligands but also function as general stress sensors via monitoring of cytosolic generation of the endogenous metabolite S1P. Our study establishes a novel activation mechanism of mammalian NOD1/2 in which these pattern recognition receptors recognize cytosolic S1P upon various stress stimuli. Recent findings have established the concept that the metabolic status of immune cells controls their responses. Most of these studies focused on glycolysis and TCA cycle, amino acid metabolism, and fatty acid synthesis (O'Neill & Pearce, 2016; O'Neill *et al*, 2016). Our findings further extend this concept by establishing a novel link between sphingolipid metabolism and NOD1/2 activation upon stress induction. Previous studies reported that sphingosine kinases/S1P are required for ER and mitochondrial unfolded protein responses and contribute to protection against ER, mitochondrial, and oxidative stress (Lee *et al*, 2015; Qi *et al*, 2015; Kim & Sieburth, 2018; Kim & Sieburth, 2019). SPHK2 is required for inflammatory responses in macrophages upon ureteral obstruction and titanium particle stimulation (Ghosh *et al*, 2018; Yang *et al*, 2018). It inhibits IL6 production at late time points upon LPS stimulation (Weigert

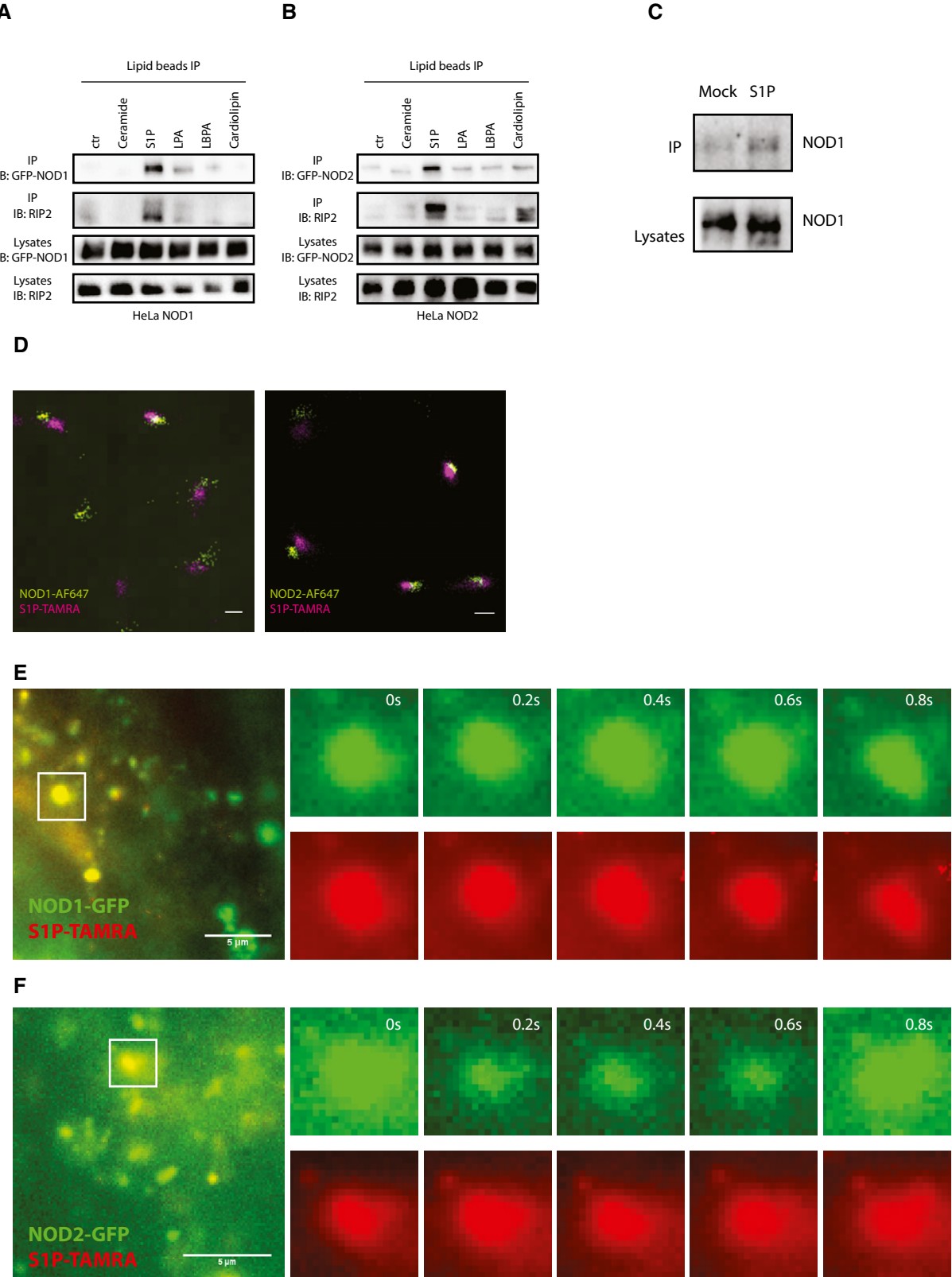

**Figure 5.**

◄

**Figure 5.  S1P interacts with NOD1/2.**

A, B   Immunoprecipitation with various lipid-coated beads using HeLa NOD1-GFP (A) and NOD2-GFP cells (B). Cell lysates of HeLa NOD1 or HeLa NOD2 cells were incubated with lipid-coated beads at room temperature for 2 h.
C       Immunoprecipitation with S1P-coated beads using THP-1 lysates.
D       Dual-color dSTORM imaging of HeLa NOD1 or NOD2 upon S1P stimulation. After stimulation with S1P-TAMRA (20 μM) for 2 h, HeLa NOD1 or NOD2 cells were fixed, permeabilized, and stained with anti-GFP antibody and Alex Fluor 647-conjugated secondary antibody. Images were reconstructed from 10,000 raw frames. Scale bar: 200 nm.
E, F   Single particle tracking of NOD1-GFP (E) or NOD2-GFP (F) with S1P-TAMRA. After doxycycline induction overnight, HeLa NOD1 or NOD2 cells were stimulated with S1P-TAMRA (20 μM) for 1 h together with digitonin. Scale bar: 5 μm.

Source data are available online for this figure.

et al, 2019). However, SPHK2 does not affect inflammatory responses upon short-time stimulation with LPS (Xiong et al, 2013). Our studies complement the current knowledge about immune roles of SPHKs by showing that S1P, a product of SPHKs, modulates cytokine responses during stress-induced inflammation. We propose that S1P may function as a central player in coordinating cellular stress responses and inflammation upon stress induction. Interaction partners and downstream effects of S1P may be context-dependent and deserve further investigations.

As a pleiotropic second messenger, S1P acts both extracellularly and intracellularly to regulate diverse processes, including immune cell trafficking, inflammation, and apoptosis (Spiegel & Milstien, 2011; Maceyka et al, 2012). Our study identifies NOD1/2 as novel receptors for cytosolic S1P, in addition to HDAC1/2 and TRAF2, which have been previously identified as cytosolic interactors (Hait et al, 2009; Alvarez et al, 2010), and establishes S1P as the missing link between cellular stress and NOD1/2-mediated inflammation. Considering that S1P is structurally and metabolically conserved through evolution (Hannun & Obeid, 2008), we propose that cytosolic S1P generated upon perturbation of cellular homeostasis represents an endogenous stress-associated molecular pattern (SAMP). Contrary to canonical damage- or danger-associated molecular patterns (DAMPs) released after cell lysis (Matzinger, 1994), S1P is generated in the cytosol upon cellular stress induction without considerable cell death. The enzymes generating S1P show distinct subcellular compartmentalization during stress responses. SPHK1 is activated and targeted to mitochondria upon mitochondrial stress (Kim & Sieburth, 2018), and SPHK2 is translocated into the ER upon serum starvation (Maceyka et al, 2005). In this study, we demonstrate that the total abundance of S1P increases up to 3 μM upon delivery of various stressors. Considering that SPHKs are enriched in intracellular compartments, we postulate that the intracellular concentration of S1P at various subcellular sites may exceed the

overall levels that we measured. Thus, it is conceivable that endogenous S1P is sufficient to induce NOD1/2 activation. How S1P generation is regulated at the spatial level and whether these subcellular sites of S1P serve as signaling hubs for cell stress responses and inflammation remain to be established. Moreover, the cellular origin, e.g., hematopoietic versus non-hematopoietic, and the cellular activation, e.g., by cytokines and microenvironment, alter the expression levels of NOD1/2 and their downstream signaling molecules. Accordingly, S1P-triggered inflammatory responses may differ in distinct tissues and at particular stages of a disease.

NOD1/2 trigger immune responses to diverse peptidoglycan-free microbes and underlying mechanisms are ill defined. We discovered that various stress stimuli induced production of the cellular metabolite S1P and subsequent NOD1/2-dependent inflammation. Since bacteria and viruses induce ER stress (Celli & Tsolis, 2015), DNA damage (Zgur-Bertok, 2013), and protein translation block (Fontana et al, 2011; Dunbar et al, 2012), it is tempting to assume that NOD1/2 sensing of host cytosolic S1P represents an ancient alert mechanism for infectious insult. Many intracellular pathogens, such as *Legionella pneumophila* and *Burkholderia pseudomallei,* secrete S1P lyases as virulence factors facilitating intracellular survival (Rolando et al, 2016; McLean et al, 2017), supporting the importance of S1P sensing by NOD1/2 during intracellular infection. S1P binds to the NBDs and activates RIP2-mediated signaling, which differs from peptidoglycan sensing via LRR domains of NOD1/2 and indicates a different activation mechanism of NOD1/2 by S1P. We discovered that S1P interacts with the amino acid (H517 of NOD1) that stabilizes the β-phosphate of ADP, which is critical for maintaining the inactive conformation of NOD1/2 (Zurek et al, 2012; Maekawa et al, 2016). Therefore, S1P could induce NOD1/2 activation by promoting their active conformation through replacing of ADP in the NBD region. This bears similarities with ATP and MDP binding to NOD2 (Mo et al, 2012). Our finding that S1P impairs the

**Figure 6.  S1P directly binds to NOD1/2 via the NBD and triggers their activation.**

A       Molecular docking of S1P with rabbit NOD2.
B       SDS–PAGE analysis of purified NOD1-GFP and NOD2-GFP.
C       Microscale thermophoresis (MST) analysis of direct binding of S1P to NOD1, NOD2, NLRP3, or GFP.
D       MST analysis of direct binding of NOD1 to S1P and iE-DAP.
E       MST analysis of direct binding of NOD2 with S1P and MDP.
F       ELISA analysis of IL8 in supernatants of HEK293T *NOD1/2* KO cells transfected with indicated *NOD1* or *NOD2* mutants. After 24-h transfection, cells were stimulated with S1P (20 μM), iE-DAP (20 μM), or MDP (20 μM) together with digitonin for 20 h.
G, H   MST analysis of direct binding of NOD1 (G) or NOD2 (H) with iE-DAP or MDP in the presence of S1P (2 μM).

Data information: (C) Means ± SD from three independent experiments. (D, E, G, H) One representative experiment out of three independent experiments. (F) Means ± SD of three technical replicates from one representative experiment out of three independent experiments. *P* values were calculated using two-way ANOVA.
*$P \le 0.05$ and ****$P \le 0.0001$.

►

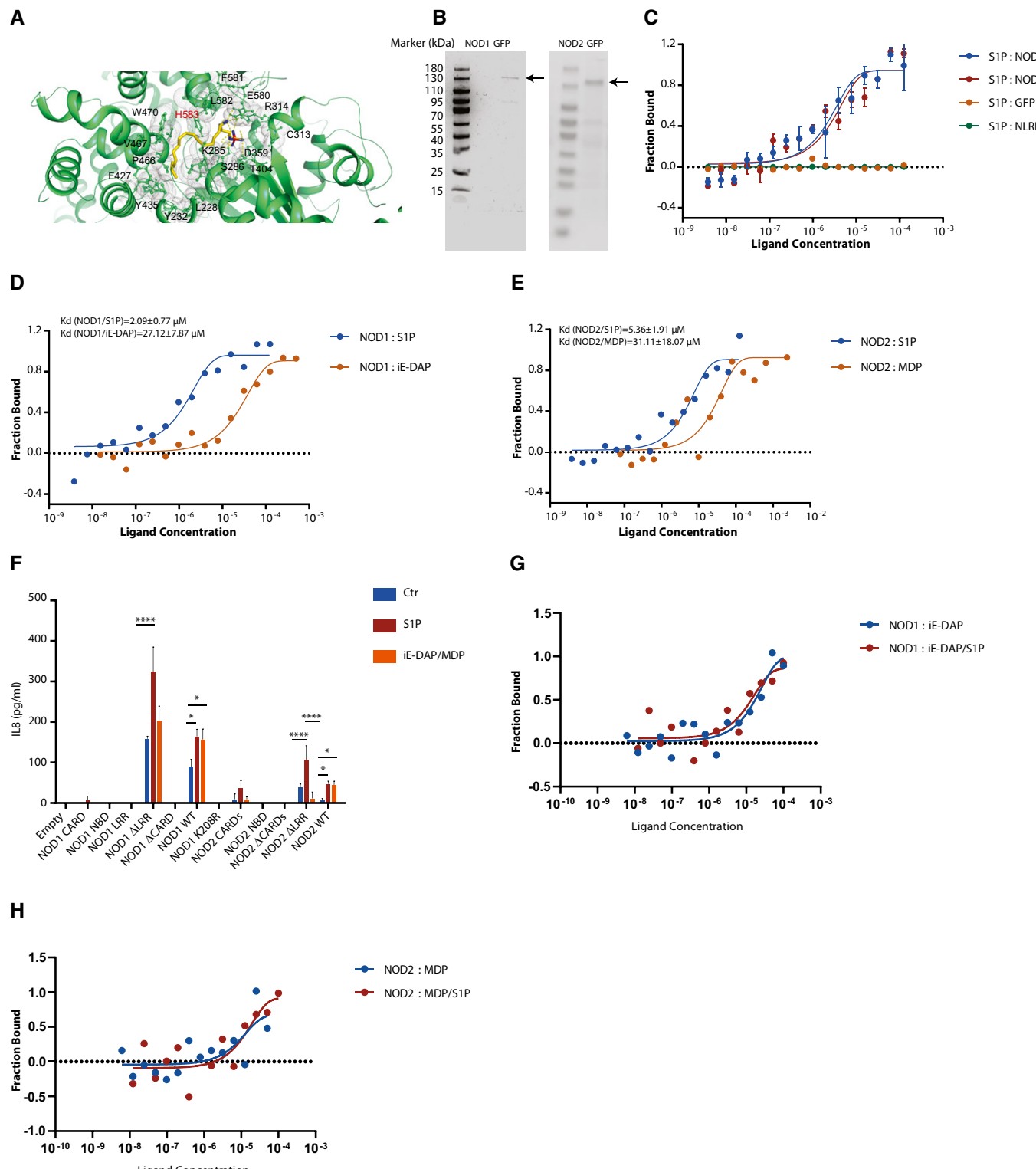

**Figure 6.**

binding of ADP to NOD1/2 supports this possibility. Further, ADP replacement stabilizes the active conformation resulting in the activation of NOD1/2. The detailed structural requirements for S1P and iE-DAP/MDP driven NOD1/2 signaling deserve further elucidation.

In context of infection, the assumption of an evolutionary conserved role of NOD1/2 in detecting pathogens or pathogen-induced cellular stress via NBDs is supported by the finding that the 290 NOD-like receptors (NLRs) identified in *Hydra magnipapillata* lack LRRs, but

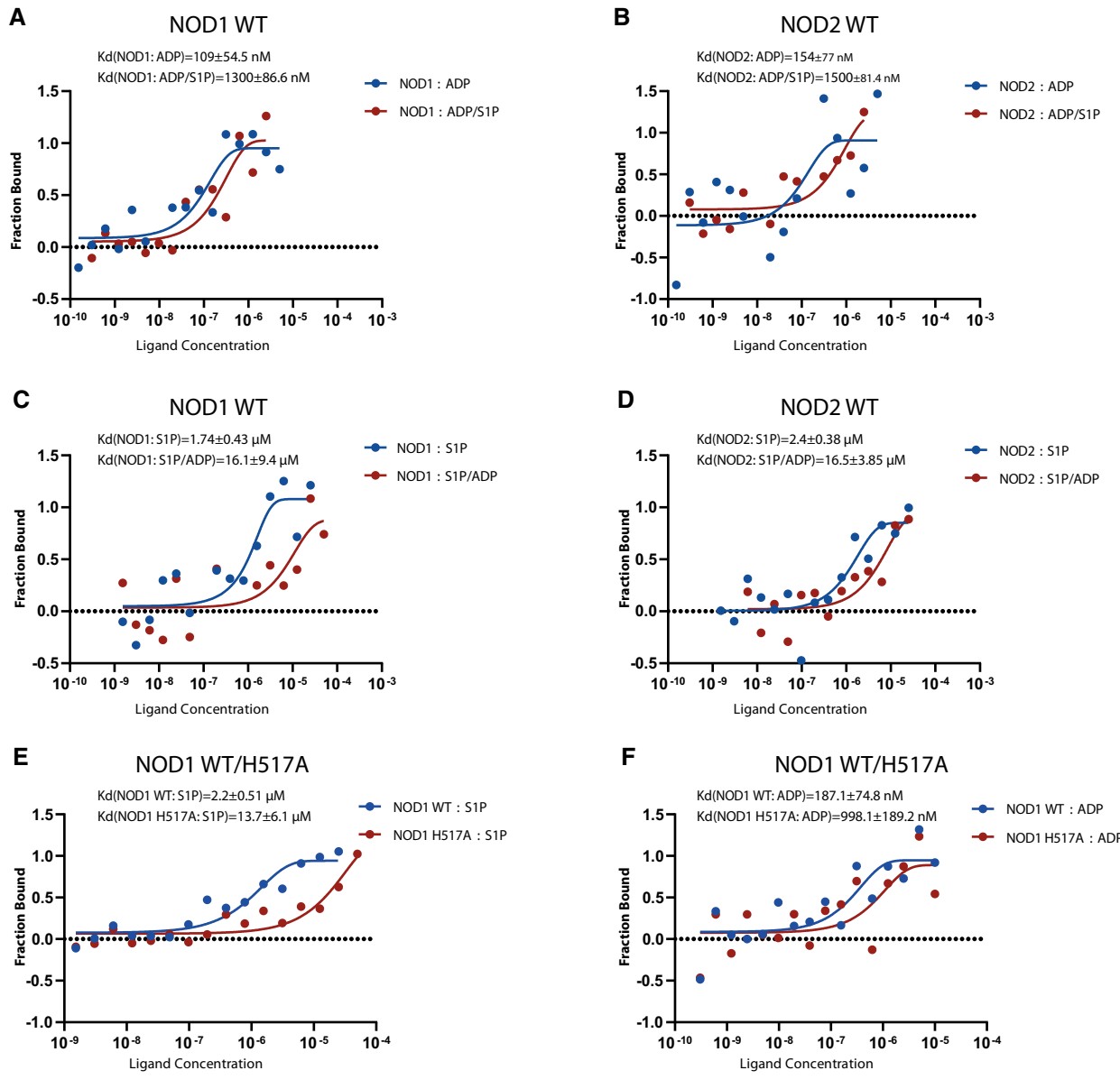

**Figure 7. S1P binds to NOD1 via H517 and impairs ADP binding.**

A, B  MST analysis of direct binding of NOD1 (A) or NOD2 (B) with ADP in the presence of S1P (2 µM).

C, D  MST analysis of direct binding of NOD1 (C) or NOD2 (D) with S1P in the presence of ADP (2 µM).

E, F  MST analysis of direct binding of S1P (E) or ADP (F) with NOD1 WT or NOD1 H517A.

Data information: (A-F) One representative experiment out of three independent experiments.

still interact with the death domain (DD) or caspase recruitment domain (CARD)-containing proteins, resembling inflammasomes in vertebrates (Lange *et al*, 2011). This together with our evidence on NOD1/2 activation by S1P via the NBDs substantiates the potential relevance of metabolites as primordial signals for infection.

Metabolites have recently gained attention for their potential roles in the host–microbiome cross talk. Most microbes belonging to the *Bacteroidetes* phylum, a dominant proportion of human gut microbiome, have been found to produce sphingolipids (Heaver *et al*, 2018). Recently, it has been established that the increased

abundance of host sphingolipids represents the most significant metabolite signature in inflammatory bowel disease (IBD) patients (Brown *et al*, 2019). Moreover, numerous NOD2 mutations in the NBD are associated with IBD, Blau syndrome, and early-onset sarcoidosis (McGovern *et al*, 2001; Caso *et al*, 2015). It has been shown that the association of plasma membrane of NOD2 is essential for MDP-induced NF-kB activation (Barnich *et al*, 2005; Lecine *et al*, 2007). Intriguingly, our data demonstrate that Crohn's disease-associated mutation NOD2 1007fs which is unresponsive to MDP is still activated by S1P. Consistently, NOD1 H517A that mainly

resides in the cytosol (Zurek *et al*, 2012) is activated by cytosolic delivery of iE-DAP with digitonin. Thus, it is plausible that cytosolic NOD1/2 can detect their ligands in the cytosol. Our findings encourage further investigations into the S1P-NOD1/2 axis in sphingolipid-mediated host–microbiome interactions under physiologic and pathologic conditions.

Non-resolving chronic inflammation resulting from the failure to reestablish homeostasis underlies the pathogenesis of many stress-related diseases, such as obesity, diabetes, and cancer (Chovatiya & Medzhitov, 2014; Kotas & Medzhitov, 2015). Hence, our findings shed light on novel molecular mechanisms of the inflammation associated with chronic diseases caused by disturbed homeostasis and establish the SPHKs-S1P-NOD1/2 axis as a potential target for novel therapeutic strategies for chronic inflammatory diseases.

# Materials and Methods

## Mice

B6N.129S6-Sphk1$^{tm1Rlp}$/J (*Sphk1$^{-/-}$*) and B6N.129S6-Sphk2$^{tm1Rlp}$/J (*Sphk2$^{-/-}$*) mice were originally obtained from the Jackson Laboratory (Bar Harbor, ME, USA). These mice were maintained under specific pathogen-free (SPF) conditions at the Max Planck Institute for Infection Biology in Berlin, Germany. 6- to 15-week-old female mice were employed for the experiments.

## Plasmids, antibodies, and reagents

NLRP3-GFP (Addgene, 73955) was obtained from Addgene. NOD1- or NOD2-related plasmids were constructed as described (Kufer *et al*, 2008). Chemicals used in this study include NVP231 (10 μM, Tocris, 3960), N-oleoylethanolamine (NOE, 30 μg/ml, Sigma, O0383), SKI-II (25 μM, Sigma, S5696), PF543 (20 μM, Sigma, PZ0234), ceranib-2 (10 μM, Sigma, SML0607), fumonisin B1 (10 μM, Sigma, F1147), deoxynojirimycin (20 μM, Sigma, D9305), GW4869 (20 μM, Sigma, D1692), imipramine (50 μM, Sigma, I0899), L-Cycloserine (100 μM, Sigma, C1159), KIRA6 (5 μM, Merck Millipore, 532281), 4μ8C (1 μM, Merck Millipore, 412512), PERK inhibitor I (10 μM, Merck Millipore, 516535), thapsigargin (5 μM, Sigma, SML1845), tunicamycin (5 μg/ml, Sigma, T7765), nigericin (5 μM, Sigma, N7143), paclitaxel (5 μM, Sigma, N7191), cytochalasin D (5 μM, Sigma, C8273), jasplakinolide (0.2 μM, Sigma, J4580), nocodazole (10 μM, Sigma, M1404), colchicine (10 μM, Sigma, C9754), vinblastine (10 μM, Sigma, V1377), monensin (Sigma, 30552), brefeldin A (0.5 μg/ml, Sigma, B6542), cycloheximide (50 μM, Sigma, C1988), anisomycin (1 μM, Sigma, A9789), actinonin (50 μM, Sigma, A6671), lovastatin (30 μM, Sigma, 1370600), carbonyl cyanide m-chlorophenyl hydrazine (CCCP, 5 μM, Sigma, C2759), etoposide (10 μM, Sigma, E1383), iE-DAP (Invivogen, tlrl-dap), MDP (Invivogen, tlrl-mdp), C2 ceramide (Enzo life sciences, BML-SL100-0005), C16 ceramide (Enzo life sciences, BML-SL115-0005), sphingosine (Enzo life sciences, BML-EI155-0025), sphingosine-1-phosphate (S1P, Enzo life sciences, BML-SL140-0001), and S1P-TAMRA (Echelon Biosciences, S-200T). S1P-coated beads (Echelon Biosciences, S-6110-2) and sphingolipid-coated bead pack (Echelon Biosciences, P-B00SS), C2 ceramide, C16 ceramide, and sphingosine, were dissolved in ethanol. S1P was

dissolved in water with sonication (stock concentration 2 mM). S1P-TAMRA was dissolved in methanol (stock concentration 200 μM). All other chemicals were dissolved in DMSO if not specified.

Antibodies used in this study include anti-ACTB (Sigma-Aldrich, A2228), anti-GFP (Proteintech, 66002-1-Ig), anti-GFP (ChromoTek, PABG1), anti-RIP2 (Cell Signaling Technology, #4142), anti-SPHK2 (ECM Biosciences, SP4621), anti-phospho-p38 (Cell Signaling Technology, #4511), anti-phospho-JNK (Cell Signaling Technology, #4668), anti-phospho-ERK1/2 (Cell Signaling Technology, #4370), anti-phospho-p65 (Cell Signaling Technology, #3033), and anti-NOD1 (Cell Signaling Technology, #3545).

## Cell culture

The human monocytic cell line THP-1 was obtained from the American Type Culture Collection (ATCC, TIB-202) and maintained in RPMI 1640 (Gibco, 31870) with 10% (v:v) heat-inactivated fetal bovine serum (FBS, Sigma-Aldrich, F0804), 1 mM sodium pyruvate (Gibco, 11360070), 2 mM L-glutamine (Gibco, 25030081), 10 mM HEPES buffer (Gibco, 15630080), pH 7.2-7.5, and 50 μM 2-mercaptoethanol (Gibco, 31350010). To differentiate THP-1 into macrophage-like cells, THP-1 cells were stimulated with 50 ng/ml phorbol 12-myristate 13-acetate (Sigma-Aldrich, P8139) for 24 h and then incubated with fresh medium for another 48 h. HEK293T WT and HEK293T NOD1/2 dKO cells were generated as described (Dagil *et al*, 2016). HeLa inducible NOD1-GFP and NOD2-GFP cells were generated by co-transfection of pcDNA5/FRT/TO-EGFP-NOD1 or pcDNA5/FRT/TO-EGFP-NOD2 (Addgene #131206 and #131207) together with pOG44 in a 9:1 ratio into HeLa FlpIN T-REx cells (kindly provided by the Hentze Lab, EMBL, Heidelberg) using Lipofectamine 2000 (Thermo Fisher Scientific) and selected with 10 μg/ml blasticidin and 500 μg/ml hygromycin B. HeLa inducible NOD1-GFP and NOD2-GFP cells, human neonatal dermal fibroblasts, and HEK293T cells (DSMZ, ACC305) were maintained in complete Dulbecco's modified Eagle's medium (DMEM) 4.5 g/l glucose (Gibco, 10938) with 10% (v:v) heat-inactivated FBS, 1 mM sodium pyruvate, and 2 mM L-glutamine. To induce NOD1 or NOD2 expression, HeLa NOD1 or NOD2 cells were treated with 0.5 μg/ml or 0.25 μg/ml doxycycline for at least 16 h, respectively. All cells have been checked for mycoplasma contamination regularly. Bone marrow-derived macrophages (BMDMs) were obtained from tibial and femoral bones and generated with DMEM containing 20% L929 cell supernatant, 10% FBS, 5% heat-inactivated horse serum, 1 mM sodium pyruvate, and 2 mM L-glutamine. For BMDM stimulation experiments, BMDMs were treated with IFN-γ (20 ng/ml) overnight before stimulation.

## Cell viability analysis

Cell viability assays were performed with CellTiter 96 AQ$_{ueous}$ one solution reagent (Promega) according to the manufacturer's instructions. Cell viability upon various stimulations was normalized to corresponding controls.

## Shigella infection

*Shigella flexneri* M90T afaE (Clerc & Sansonetti, 1987) was kindly provided by Philippe Sansonetti (Institute Pasteur) and were grown

in Caso Broth containing 200 µg/ml spectinomycin. HeLa cells were infected at multiplicity of infection (MOI) of 10 in DMEM without supplements. After 15 min of sedimentation at room temperature, infection was carried out at 37°C and 5% $CO_2$. After 30 min, the medium was replaced with 250 µl DMEM containing 100 µg/ml gentamycin. Supernatants were collected at the indicated time points postinfection for IL8 measurement with ELISA kits (Bio-Techne, DY208).

## NF-κB Luciferase assay

HEK293T WT and *NOD1/2* KO NF-κB reporter cells were seeded in 96-well plates and stimulated with various stressors for 8 h. Then, cells were lysed in lysis buffer (Thermo Fisher Scientific) and luciferase activities were determined with firefly luciferase glow kit (Thermo Fisher Scientific) according to the manufacturer's instructions. Luciferase activities were normalized to the amount of protein determined with Coomassie Plus kit (Thermo Fisher Scientific). Fold inductions were calculated against normalized luciferase activities of corresponding controls.

## Lipidomic profiling and analysis of sphingosine-related metabolites

For lipidomic profiling, $1 \times 10^6$ cells were treated with indicated stimuli for 2 h. After washing twice with Dulbecco's phosphate-buffered saline (DPBS) without calcium and magnesium, cells were collected and suspended in 300 µl of DPBS. Afterward, cells were transferred into 1.5-ml Eppendorf tubes precooled at −80°C. The lipidomic analysis was performed by Lipotype GmbH. For analysis of sphingosine-related metabolites, $3 \times 10^6$ cells were seeded in Lumox 50 dishes and treated with indicated stimuli for 2 h. After washing twice with 0.9% NaCl preconditioned at 37°C, the membrane of Lumox dishes was placed into tubes precooled at −80°C. 600 µl of quenching solution (dichloromethane/ethanol) was added into each tube. The measurement was performed by Metanomics Health GmbH.

## S1P quantification by ELISA

Quantification of S1P abundances in cell lysates was performed with S1P ELISA kit (K-1900, Echelon Biosciences) according to the manufacturer's instructions with some modifications. Human neonatal dermal fibroblasts were grown in DMEM full medium. For each condition, 2x T75 flask cells were required. After 4-h stimulation with various stressors, cells were washed with PBS and collected in lysis buffer (20 mM PIPES PH7.4, 150 mM NaCl, 1 mM EGTA, 1% v/v Triton X-100, 1.5 mM $MgCl_2$, 0.1% SDS, 1X PhosSTOP, 1X Protease inhibitor cocktail without EDTA). Protein concentration was measured by BCA assay kit (Thermo Fisher) and diluted with delipidized serum to 1 µg/µl. Then, the diluted samples were added into S1P ELISA plates according to the instructions.

## Bioplex analysis

Bioplex analysis of supernatants of BMDMs was performed using Bio-Plex Pro Mouse chemokine panel 33-plex and human cytokine 27-plex (Bio-Rad) in a Bio-Plex 200 system according to the manufacturer's instructions.

## Site-directed mutagenesis

The mutagenesis was performed with Q5® site-directed mutagenesis kit (E0554, NEB) according to the manufacturer's instructions. The primers used in this study were as follows: NOD1 E157A F: 5′-GCTGCTGGAGgcgATCTACATGG-3′, NOD1 E157A R: 5′-AGCTCCTCCTTCTGGGCA-3′, NOD1 D161A F: 5′-GATCTACATGgcgACCATCATGGAGCTGG-3′, NOD1 D161A R: 5′-TCCTCCAGCAGCAGCTCC-3′, NOD1 D203A F: 5′-CATCCTGGGTgcgGCTGGGGTGG-3′, D203A R: 5′-AAGATGGTCTCACCCTGC-3′, L218A F: 5′-GCTGCAGAGCgcgTGGGCCACGG-3′, L218A R: 5′-CGCTGTAGCAGCATGGAC-3′, R237A F: 5′-CTTTCGCTGCgcgATGTTCAGCTG-3′, R237A R: 5′-TGGAAGAAGAATTTGACC-3′, H257A F: 5′-GCTCTTCAAGgcgTACTGCTACCCAGAGCGG-3′, H257A R: 5′-AGGTCCTGCAGACACAGC-3′, E267A F: 5′-GGACCCCGAGgcgGTGTTTGCCT-3′, E267A R: 5′-CGCTCTGGGTAGCAGTAGTG-3′, H290A F: 5′-GGACGAGCTGgcgTCGGACTTGGACC-3′, H290A R: 5′-AGGCCATCGAAGGTGAAG-3′, E306A F: 5′-CTGCCCCTGGgcgCCTGCCCACC-3′, E306A R: 5′-GAGCTGTCAGGCACGCGGC-3′, K324A F: 5′-GAAGCTGCTCgcgGGGGCTAGCAAG-3′, K324A R: 5′-CCACTGAGCAGGTTGGCC-3′, K328A F: 5′-GGGGGCTAGCgcgCTGCTCACAG-3′, K328A R: 5′-TTGAGCAGCTTCCCACTG-3′, R340A F: 5′-CGAGGTCCCGgcgCAGTTCCTGCGGAAG-3′, R340A R: 5′-ATGCCTGTGCGGGCTGTG-3′, R344A F: 5′-CCAGTTCCTGgcgAAGAAGGTGCTTCTCCGG-3′, R344A R: 5′-CGCGGGACCTCGATGCCT-3′, R350A F: 5′-GGTGCTTCTCgcgGGCTTCTCCC-3′, R350A R: 5′-TTCTTCCGCAGGAACTGG-3′, D372A F: 5′-GGCCCTGCAGgcgCGCCTGCTGA-3′, D372A R: 5′-CGCTCGGGGAACATCCTC-3′, R373A F: 5′-CCTGCAGGACgcgCTGCTGAGCCAGCTG-3′, R373A R: 5′-GCCCGCTCGGGGAACATC-3′, R399A F: 5′-GATCATCTTCgcgTGCTTCCAGCACTTCC-3′, R399A R: 5′-CAGCAGAAGAGGGGCACA-3′, D423A F: 5′-GACCCTGACAgcgGTCTTCCTCCTGGTC-3′, D423A R: 5′-ATCGTGCAGTCGGGCAGC-3′, R435A F: 5′-5′-CCATCTGAACgcgATGCAGCCCAGC-3′, R435A R: 5′-ACCTCAGTGACCAGGAGG-3′, Q484A F: 5′-GGAGGAGGTGgcgGCCTCCGGGC-3′, Q484A R: 5′-TGGGTGAAGACAAAGAGGCTC-3′, L495A F: 5′-AGACATGCAGgcgGGCTTCCTGCGG-3′, L495A R: 5′-CTCTCCTGCAGCCCGGAG-3′, E514A F: 5′-GCAGTCCTATgcgTTTTTCCACCTCACCCTCCAGG-3′, E514A R: 5′-TGGTCACCCCCGGGGCCC-3′, H517A F: 5′-TGAGTTTTTTgcgCTCACCCTCCAGGCCTTCTTTAC-3′, H517A R: 5′-TAGGACTGCTGGTCACCC-3′, L540A F: 5′-CACTCAGGAGgcgCTCAGGTTCTTCCAGG-3′, L540A R: 5′-CCCACCCTGTCGTCCAGC-3′, F544A F: 5′-GCTCAGGTTCgcgCAGGAGTGGATGCCC-3′, and F544A R: 5′-AGCTCCTGAGTGCCCACC-3′.

## Purification of NOD1-GFP and NOD2-GFP

HeLa NOD1 or NOD2 cells from 20 T175 flasks were collected and lysed with 5 ml lysis buffer (50 mM Tris–HCl, pH 7.5, 150 mM NaCl, 1% NP-40, 10 mM DTT, 1 mM EDTA supplemented with proteases inhibitor cocktail). The lysates were incubated with 2 ml GFP-Trap agarose at 4°C for 4 h. The agarose was collected with Poly-Prep columns by gravity flow. Afterward, the agarose was sequentially washed with 50 ml wash buffer I (50 mM Tris–HCl, pH 7.5, 150 mM NaCl) and 50 ml wash buffer II (50 mM Tris–HCl, pH 7.5, 300 mM NaCl). To remove heat shock proteins, columns were further washed with 50 ml wash buffer III (50 mM Tris–HCl, pH 7.5, 150 mM NaCl, 5 mM $MgCl_{2+}$, 1 mM ATP). To elute the proteins, the agarose was mixed with 2 ml elution buffer (100 mM Glycine, pH 2.5) for 1 min and the

eluted proteins were immediately neutralized with 0.2 ml of 1 M Tris, pH 10.5.

## Dual-color direct stochastic optical reconstruction microscopy (dSTORM) and single particle tracking in live cells

For dSTORM, HeLa inducible NOD1-GFP or NOD2-GFP cells were grown on μ-Slides with glass bottom (ibidi, Germany) and treated with doxycycline overnight to induce NOD1-GFP or NOD2-GFP expression. Afterward, cells were stimulated with S1P-TAMRA (20 μM) in the presence of digitonin (5 μg/ml) for 2 h. Then, cells were fixed with 4% (v:v) paraformaldehyde in PBS, pH 7.4 for 10 min at room temperature (RT). After washing twice with PBS, cells were incubated with 50 mM glycine in PBS, pH 7.4 for 10 min and permeabilized with 0.05% saponin (Sigma-Aldrich, 47036), 1% BSA in PBS for 10 min. Rabbit anti-GFP antibody (ChromoTek) and Alex Fluor 647-conjugated goad anti-rabbit antibody (Thermo Fisher Scientific) were diluted in PBS and incubated for 1 h at room temperature. The super resolution imaging was performed with Nanoimager S (Oxford Nanoimaging). For single particle tracking in live cells, Nanoimager S was pre-warmed at 37°C. HeLa inducible NOD1-GFP or NOD2-GFP cells were stimulated with S1P-TAMRA (20 μM) in the presence of digitonin (5 μg/ml) for 30 min. Medium was then replaced with fresh DMEM without phenol red. Live imaging was performed with Nanoimager S at acquisition speed of 100fps.

## Gene expression analysis by real-time quantitative reverse-transcription PCR (qRT–PCR) and Fluidigm

Total RNA was isolated with TRIzol reagent, as described by the manufacturer (Invitrogen). RNA (1 μg) was used to generate cDNA via the iScript cDNA Synthesis Kit (Bio-Rad), and qRT–PCR was performed using Power SYBR Green Master Mix (Applied Biosystems) in a StepOne Plus thermocycler (Applied Biosystems). The average threshold cycle (Ct) of quadruplicate reactions was employed for all subsequent calculations using the ΔΔCt method. Gene expression was normalized to glyceraldehyde-3-phosphate dehydrogenase (GAPDH), and fold changes were calculated against corresponding controls. qRT–PCR data were representative of at least three independent experiments, with at least 2 technical replicates per experiment. The sequences of primers used in this study were as follows: hGAPDH F: 5′-GGAGCGAGATCCCTCCAAAAT-3′, hGAPDH R: 5′-GGCTGTTGTCATACTTCTCATGG-3′, hIL6 F: 5′-AC TCACCTCTTCAGAACGAATTG-3′, hIL6 R: 5′-CCATCTTTGGAAGGT TCAGGTTG-3′, hIL8 F: 5′-TTTTGCCAAGGAGTGCTAAAGA-3′, hIL8 R: 5′-AACCCTCTGCACCCAGTTTTC-3′, mGAPDH F: 5′-AGGTCGG TGTGAACGGATTTG-3′, mGAPDH R: 5′-TGTAGACCATGTAGTTG AGGTCA-3′, mIL6 F: 5′-TAGTCCTTCCTACCCCAATTTCC-3′, mIL-6 F: 5′-TTGGTCCTTAGCCACTCCTTC-3′, hS1PR1 F: 5′-TTCCACCG ACCCATGTACTAT-3′, hS1PR1 R: 5′-GCGAGGAGACTGAACACGG-3′, hS1PR2 F: 5′-CTAGCCAGTTCTGAAAGC-3′, hS1PR2 R: 5′-ACAG AGGATGACGATGAAG-3′, hS1PR3 F: 5′-GAGGAGCCCTTTTTCAAC-3′, hS1PR3 R: 5′-TCATTTCAAAGGGAAGCG-3′, hS1PR4 F: 5′-GAC GCTGGGTCTACTATTGCC-3′, hS1PR4 R: 5′-CCTCCCGTAGGAAC CACTG-3′, hS1PR5 F: 5′-AGGAAGCTCAGTTCACAG-3′, and hS1PR5 R: 5′-GATTCTCTAGCACGATGAAG-3′.

Gene expression was analyzed simultaneously with the 96.96 Dynamic Array Integrated Fluidic Circuits from Fluidigm as

previously described (Lozza et al, 2014). Preamplification of genes by reverse transcription and cDNA synthesis (18 cycles) was performed with the Cells Direct one-Step qPCR kit (Life Technologies, Inc.) and TaqMan gene expression assay mix (Applied Biosystems). Gene expression was normalized to glyceraldehyde-3-phosphate dehydrogenase (GAPDH). Data represent fold changes ($2^{-\Delta\Delta CT}$) in transcripts relative to the appropriate internal control (DMSO). Data were generated from 2 technical replicates and at least three independent experiments. TaqMan probes are listed in Table S1.

## Microarray analysis

Total RNA was isolated with TRIzol (Invitrogen) following the manufacturer's protocol using glycogen as co-precipitant. Quality control and quantification of total RNA was analyzed using an Agilent 2100 Bioanalyzer (Agilent Technologies) and a NanoDrop 1000 UV–Vis spectrophotometer (Thermo Fisher Scientific). Microarray experiments were performed as single-color hybridization. Total RNA was amplified and labeled with the Low Input Quick-Amp Labeling Kit (Agilent Technologies). In brief, mRNA was reverse-transcribed and amplified using an oligo-dT-T7 promoter primer and labeled with cyanine 3-CTP. After precipitation, purification, and quantification, 0.75 μg labeled cRNA was fragmented and hybridized to custom whole genome human 8 × 60K multipack microarrays (Agilent-048908) according to the supplier's protocol (Agilent Technologies). Scanning of microarrays was performed with 3 μm resolution (8x60K) using a G2565CA high-resolution laser microarray scanner (Agilent Technologies). Microarray image data were processed with the Image Analysis/Feature Extraction software G2567AA v. A.11.5.1.1 (Agilent Technologies) using default settings and the GE1_1105_Oct12 extraction protocol. Expression data were analyzed using R scripts. The data were quantile-normalized between arrays, and batch effect of purification state was removed by ComBat (Johnson et al, 2007; Leek et al, 2012). The differentially expressed genes were assessed using limma (Smyth, 2005). Genes with corrected P-values of < 0.05 were considered significant after Benjamini–Hochberg correction for multiple testing. Functional enrichment was analyzed by CERNO method implemented in the tmod R package (Zyla et al, 2019) with genes sorted by maximum significance difference (Zyla et al, 2017). As pathway collection, the Hallmark MSigDB (Liberzon et al, 2011; Liberzon et al, 2015) was used. The pathways with P-values of < 0.01 after Benjamini–Hochberg correction for multiple testing were considered as significant. For the most differentially expressed pathways, their eigengene vector of PCA first component across samples was calculated. All analyses were conducted in R, and all scripts are available upon request.

## Immunoprecipitation (IP)

HeLa NOD1 or NOD2 cells were grown overnight and afterward treated with doxycycline overnight to induce NOD1 or NOD2 expression. Cells of one 10-cm dish were lysed with 500 μl of lysis buffer (25 mM Tris–HCl, pH 7.5, 150 mM NaCl, 1 mM EDTA, 1% NP-40, and 5% glycerol) supplemented with protease inhibitors (Roche, 05892791001) on ice for 20 min. Lysates were centrifuged at 16,000 × g at 4°C for 10 min, and supernatants were mixed with

GFP-Trap agarose (ChromoTek GmbH, gta-20) overnight at 4°C. Afterward, agaroses were washed twice with PBS and boiled with sample buffer for SDS–PAGE. For lipid-coated beads IP, lipid-coated beads were incubated with supernatants for 2 h at room temperature and then washed with lysis buffer for three times and boiled with sample buffer for SDS–PAGE.

### SDS–PAGE and Western blot

Cells were lysed in RIPA buffer supplemented with protease inhibitor cocktail and PhosSTOP (Roche, 4906837001) on ice for 10 min. After centrifugation at $16,000 \times g$ for 10 min, supernatants were heated with SDS sample buffer at 95°C for 10 min. Proteins were separated on 4–15% SDS gels (Bio-Rad, 4561086) and transferred onto nitrocellulose membranes. The membranes were incubated with primary antibodies at 4°C overnight and secondary antibodies at RT for 1 h, respectively. All antibodies were diluted in PBS supplemented with 0.1% Tween-20 (Sigma-Aldrich, P1379) and 5% BSA. Membranes were developed with ECL detection Kit (Thermo Fisher Scientific, 34094) and exposed with ChemiDoc (Bio-Rad). The antibody against ACTB/β-actin was used as loading control.

### Microscale thermophoresis analysis

Binding of NOD1, NOD2, GFP, or NLRP3-GFP to different ligands was measured by microscale thermophoresis (MST). 20 nM GFP or GFP-tagged proteins in MST buffer (PBS pH 7.4, containing 0.5% NP-40 and 10 mM DTT) was incubated with different concentrations of ligands. Immediately, samples were loaded into standard glass capillaries (NanoTemper) and thermophoresis analysis was performed on a NanoTemper Monolith NT.115 instrument (40% LED, 80% MST power) at 22°C. A laser on-time of 30 s and a laser off-time of 5 s were used. The experiment was performed in triplicates, and the MST curves were fitted using NT analysis software to obtain the $K_d$ values.

### Molecular docking

AutoDock Tools (Morris et al, 2009) were used to perform blind docking of the substrate S1P to NOD2 protein. The structure of NOD2 was obtained through the PDB with entry ID 5IRN (Maekawa et al, 2016). The structure of S1P was obtained through the PDB with entry ID 2YG2 (Christoffersen et al, 2011). Preparation of molecules included removal of water and ADP molecules, addition of all hydrogen, computation of Gasteiger charges, and merging of nonpolar hydrogen. The grid dimensions were chosen to explore the entire structure. The output complex structures were generated with the program Pymol. The final docking conformation was chosen based on the lowest binding energies. The force field of S1P and MD simulation of 10 ns were achieved by AMBER 18 (Case et al, 2005) to validate the docked result.

### Molecular dynamics (MD) simulation

10-ns MD simulations were performed to validate the docked results. The docked results were used as the initial structures to run the MD simulations. The AMBER99SB-ILDN force field (Lindorff-Larsen et al, 2010) and the general Amber force field (gaff) (Wang et al, 2004) were utilized for the protein and ligand, respectively. The atom partial charges parameters and the missing force field parameters for S1P were created by using the antechamber program (Wang et al, 2006). Nine sodium atoms were added as counterions to neutralize the net charge of the simulation system. TIP3P water molecules (Mark & Nilsson, 2001) were added to solvate the systems, and the solute atoms were at least 12 Å away from the box edges. The steepest decent minimization was used until the maximum iteration steps reached 20,000 or the convergence criterion (the root-mean-square of the energy gradient is $< 1.0 \times 10^{-4}$ kcal/mol·Å) was satisfied. The system was first minimized with the restraints (the force constant of 100 kcal/mol·Å$^2$) on the solute (includes the protein and ligand) followed by another round of free minimization. The systems were heated from 0 to 300 K (force constant of 10 kcal/mol·Å$^2$) on the protein and ligand in 100 ps. Then, the unrestrained systems were equilibrated for 1 ns with Langevin thermostat in the NPT (P = 1 atm and T = 300 K) ensemble (Loncharich et al, 1992). During MD simulations, the particle mesh Ewald (PME) method (Essmann et al, 1995) was adopted to deal with the long-range electrostatic interactions. The non-bonded cutoff was set to 9 Å, and all bonds involving hydrogen atoms were fixed to their equilibrium values using the SHAKE algorithm (Lambrakos et al, 1989). The production runs were performed under 300 K with a time step of 2 fs. The MD production runs were carried out by CUDA-version Amber16 (Gotz et al, 2012) in the NPT ensemble.

### Statistical analysis

Statistical analysis was performed with GraphPad Prism v7.03 (GraphPad software Inc., USA). Normal distribution has been evaluated with the Shapiro–Wilk test using GraphPad Prism. To determine statistical significance among investigated groups, Student's t-test and one-way or two-way analysis of variance (ANOVA) with Holm–Sidak post hoc test and ANOVA test of linear mixed model for fix effect of main factors were performed. A two-tailed P value of < 0.05 was considered to be significant.

# Data availability

Microarray data have been deposited in the Gene Expression Omnibus (GEO; www.ncbi.nlm.nih.gov/geo/) of the National Center for Biotechnology Information and can be assessed with the GEO accession number GSE124828.

**Expanded View** for this article is available online.

### Acknowledgements

The authors thank Dr. Maximiliano Gutierrez (The Francis Crick Institute, UK) for critically reading the manuscript; Dr. Emma Hooley (Oxford Nanoimaging, UK) for the help with dSTORM and single particle tracking; Ellen Heinemann, Dagmar Oberbeck-Mueller, Silke Bandermann (Max Planck Institute for Infection Biology, Germany), and Silke Rehbein, Lisa Loerzer, Ulrike Zedler (Friedrich-Loeffler-Institut) for technical assistance; Neha Agrawal for isolation of human CD14$^+$ monocytes; and Diane Schad (Max Planck Institute for Infection Biology, Berlin, Germany) for help with graphics. This work was supported by

Deutsche Forschungsgemeinschaft (KU-1945/4-1) to T.K., ERC starting Grant (PGNfromSHAPEtoVIR 202283) to I.G.B., National Key R&D Program of China (2018YFE0202301) and National Natural Sciences Foundation of China (21904138) to L. H., intramural funding of the Max Planck Society to A.D. and S.H.E.K., and EU FP7 project "ADITEC" (HEALTH-F4-2011-280873), EU Horizon 2020 "TBVAC 2020" (Grant No. 643381), BMBF "inVAC" (Grant No. 03ZZ0806A), "BioVacSafe" (IMI JU Grant No. 115308), and "PreDiCT-TB" (IMI JU Grant No. 115337) to S.H.E.K..

## Author contributions

GP, AD, and SHEK conceived and designed the study, analyzed the data, and wrote the manuscript. GP designed and performed most of the experiments. JZ, JW, and H-JM conducted microarray and data analysis. LH performed molecular docking. MD performed molecular dynamic simulation. PM-A generated knockdown cells. PS participated in the mouse experiments. LL performed Fluidigm gene expression analysis. TAK provided cell lines, plasmids, and conceptual discussions. KE and CA generated HeLa inducible GFP, NOD1-GFP, and NOD2-GFP cells. HS performed Shigella infection experiments. NN, YD, MP, and IGB helped with the experiments. All the coauthors commented on the manuscript.

## Conflict of interest

The authors declare thet they have no conflict of interest.

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
