## [Review Process File · The EMBO Journal]

Cellular stress promotes NOD1/2-dependent inflammation via the endogenous metabolite sphingosine-1-phosphate

Gang Pei, Joanna Zyla, Lichun He, Pedro Moura-Alves, Heidrun Steinle, Philippe Saikali, Laura Lozza, Natalie Nieuwenhuizen, January Weiner, Hans-Joachim Mollenkopf, Kornelia Ellwanger, Christine Arnold, Mojie Duan, Yulia Dagil, Mikhail Pashenkov, Ivo Boneca, Thomas Kufer, Anca Dorhoi, and Stefan Kaufmann

DOI: 10.15252/embj.2020106272

Corresponding author(s): Stefan Kaufmann (kaufmann@mpiib-berlin.mpg.de) , Anca Dorhoi (anca.dorhoi@fli.de), Gang Pei (gang.pei@fli.de)

Review Timeline:

Submission Date:	17th Jul 20
Editorial Decision:	7th Sep 20
Revision Received:	15th Feb 21
Editorial Decision:	12th Mar 21
Revision Received:	25th Mar 21
Accepted:	29th Mar 21

Editor: Karin Dumstrei

Transaction Report:

Dear Stefan,

Thank you for submitting your revised manuscript to The EMBO Journal. I am sorry for the delay in getting back to you with a decision, but I have now heard back from the three referees on the study.

As you can see from the comments below, the referees find the analysis interesting but also that some further work is needed to consider publication here. The referees raise issues with the overexpression system uses and I think it would be good to further clarify the use of different cell lines and measures taken to increase NOD1/NOD2 expression. I like the suggestion by referee #2 point 1 to measure physiological S1P concentration upon different cellular stressors. The referees also point out that we need some more insight into S1P binding to NOD1/2 and how this affects downstream signaling. Should you be able to address the raised concerns then I would like to invite you to submit a revised manuscript.

I think it would be helpful to discuss the revisions and would be happy to do so either via email or a video call.

Thank you for the opportunity to consider your work for publication. I look forward to discussing the revisions further with you.

Yours sincerely,

Karin

Karin Dumstrei, PhD
Senior Editor
The EMBO Journal

- a point-by-point response to the referees' comments, with a detailed description of the changes made (as a word file).

- a word file of the manuscript text.

- individual production quality figure files (one file per figure)

- a complete author checklist, which you can download from our author guidelines

(<https://www.embopress.org/page/journal/14602075/authorguide>).

- Expanded View files (replacing Supplementary Information)

Further information is available in our Guide For Authors:

The revision must be submitted online within 90 days; please click on the link below to submit the revision online before 6th Dec 2020.

Referee #1:

This manuscript describes a potential role of NOD1/2 in sensing cellular stress via intracellular sphingosine-1-phosphate (S1P).

Although drugs employed effect different cell functions, the majority of the work was done focusing on the chemical compounds that induce ER stress or inhibition of protein synthesis. The concept that NOD1/2 are involved in sensing ER stress or protein misfolding is not novel. Nevertheless, in previous reports both processes were driven by engagement with external bacterial peptidoglycan components rather than self-components.

This whole study is based on in vitro experiments performed with cell lines or in vitro established cultures of primary cells. Moreover, endogenous NOD1/2 expression appears not to be sufficient

since the authors employed different strategies to increase their expression (e.g. HeLa cells were transfected with constructs for inducible NOD1 and NOD2 expression, BMDM were pretreated with IFN- γ). This raises a major concern regarding to what degree the findings described have any real physiological relevance.

Moreover, different aspects of this work were tested using different cell types. For example, cycloheximide is among the best inducers of IL-6 in NOD transfected HeLa cells, but MIP-2 production by BMDM is highest after tunicamycin exposure. Lipidomic profiling was performed on human primary fibroblasts, in which stimulation with cycloheximide indeed induced a lot of S1P but tunicamycin not so much. Indeed, the former is the best trigger of S1P and the latter less so.

While in this study sphk2 was required for secretion of inflammatory cytokines in response to cell stress, Sphk2 is known as a down-modulator of inflammatory responses since deficient myeloid cells display increased inflammatory cytokine production after LPS stimulation. The authors need to at least discuss this apparent discrepancy.

The characterization of S1P binding to NACHT/NBD domain of NOD1/2 that results in Rip2K activation is interesting but needs additional experiments. It is not clear if S1P binding interferes with NOD1/2 dimerization that is essential for Rip2K activation. Does it display ADP? Since ADP mediates interaction between the central nucleotide-binding domain (NBD) and the winged-helix domain (WHD) is critical for stabilizing the closed conformation of NOD1/2, is it possible that S1P binding to NOD1/2 triggers or accelerates NOD degradation? Does S1P affect binding of the respective bacterial ligands to NOD1/2?

While NOD1 is broadly expressed among hematopoietic and non-hematopoietic cells, expression of NOD2 is significantly more restricted. Thus, the concept that either has the same/equally role as a general cell stress-sensing function is somewhat surprising.

Referee #2:

In the manuscript "The endogenous metabolite sphingosine-1-phosphate (S1P) activates NOD1/2-dependent inflammation triggered by cellular stress" by Pei et al, the authors report a role of NOD1/2 in surveillance of cellular stress through direct sensing of cytosolic metabolite S1P. Previous studies have demonstrated the involvement of S1P signalling in regulating heat shock proteins and the unfolded protein response in the ER and ER stress (i.e. Park et al, 2015). Furthermore, NOD1/2 roles in detecting the cellular stress have also been reported (i.e. Keestra-Gounder et al., 2016). In this regard, this new study could potentially add to the list of the very few known S1P molecular targets (i.e. TRAF2, cIAP) and also fill in a significant gap in the cell biology of NOD1/2-dependent cellular stress response. In order to strengthen the conclusions and increase the impact of this work, some additional controls and clarifications are required, however I believe the authors are well-positioned to address the comments.

Major concerns

1) Most of the studies assessing S1P induced NOD1/2 signaling involved overexpression systems in cell lines using digitonin at a final S1P concentration of 10 μ M. For a meaningful conclusion, the authors are encouraged to measure what is physiological S1P concentration that can be achieved under various cellular stressors such as tunicamycin or thapsigargin.

2) It is now well established that NOD1/2 are peripheral membrane proteins and the endosomal membranes are the sites of peptidoglycan-induced NOD1/2-dependent signal transduction (Irving et al., 2014; Nakamura et al., 2014), which surprisingly the authors did not discuss. Particularly concerning is that their studies are done mostly using overexpressing cell lines without appropriate controls. For instance, the control would be a cell line overexpressing Crohn's disease associated NOD2 mutant (i.e. 3020insC; Nakamura et al., 2014) that is expected to be unable to respond to MDP stimulation while having an intact NACHT domain. This is also a concern because overexpression of the CARD(S) domain of NOD1 and NOD2 is sufficient to induce NF- κ B.

3) Seemingly, the predicted S1P binding site coincides with the nucleotide binding pocket (Fig 6A). Thus, the authors mention that H517 in the winged-helix (WH) domain is critical for S1P binding and function (Fig S8E). However, the same amino acid was shown to be critical for nucleotide binding and it was also shown to affect subcellular localization when mutated (Zurek et al., 2011). This needs clarification.

4) In addition, it was shown that the NOD1 affinity for ATP and ADP is in nanomolar range (IC₅₀ of ~15 nM for ATP and ~190nM for ADP)(Askari et al., 2012). The authors MST results showed that the NOD1/2 affinity for S1P is low micromolar range (Fig 6C,D and E). Competitive displacement experiments to determine if the S1P binding is affected by ATP or ADP binding are required to support the conclusions.

5) Despite the analysis performed in this study, the question remains as to whether S1P binding to NOD1/2 is specifically responsible for S1P-induced NOD1/2-dependent signaling and it's not an indirect effect (see point #1). If the authors are correct, then the authors should be able to generate a mutant NOD1/2 protein that does not bind to SP1 but it's still able to bind ATP. This mutant should be introduced into NOD-deficient cells and examined for S1P induced signaling and localization.

6) The experiments to show interaction between NOD1/2 and SP1 (Fig 5A,B) should be improved. The analysis should include analysis of wild-type and predicted point mutations of SP1 binding site with or without ATP/ADP.

Minor points

1) Did the authors observe any synergistic effect of S1P and iE-DAP/MDP co-stimulation on NOD1/2 dependent signaling?

2) Fig S1A. MWs of NOD1 and NOD2 appears to be the same.

3) I could not find details about the lipid coated beads.

4) The authors should include more detail about their molecular dynamic analysis.

Referee #3:

Pei and colleagues identified the cytosolic immune receptors for peptidoglycan fragments as novel sensors for various unrelated stressors. In this regard, Nod-activation required endogenous cytosolic sphingosine-1-phosphate production. The authors present extensive experimental work using multifarious methods.

Major concerns:

- Order and number of the used reagents differ between experiments.
- All experiments are performed under basal conditions. However, NOD-receptors have their specific immune function especially during infection. Are the identified mechanisms also relevant under these conditions? This should be investigated, at least exemplarily.
- Figure 1A,B: More or less all used stressors were effective to inhibit NOD-dependent Il6... are there stressors being not effective? Such controls are needed. Maybe TLR-agonists. Are the detected differences statistically different? As well in many other experiments, e.g. Fig. S2.
- Fig. 1: Compared to Fig. S1 order of columns is different, this is confusing.
- How often were the Western blots performed? A quantification of the blots would be helpful.

Minor concerns:

- Results page 11, below: Fig. S9B, S9C should be Fig. S8B, S8C.

Ref: EMBOJ-2020-106272

Title: The endogenous metabolite sphingosine-1-phosphate (S1P) activates NOD1/2-dependent inflammation triggered by cellular stress

We thank the reviewers for their time, thoughtful comments, and appreciate the opportunity to respond. According to the reviewers' constructive suggestions, we have measured S1P abundance upon various stimulations and demonstrated that the levels of endogenous S1P are relevant in activating NOD1/2. We have performed MST assays to confirm that S1P does not affect the binding of iE-DAP/MDP to NOD1/2. Further, S1P indeed competes with ADP for binding to NOD1/2. We also purified NOD1 H517A mutant and demonstrated that H517 is critical for the binding of S1P. Finally, we have performed experiments with Shigella infection and demonstrated that S1P is required for IL8 production in HeLa cells upon Shigella infection. Together, we believe that the criticisms and comments are all addressed in the revised manuscript (Revisions are highlighted as yellow color). Please find our point-by-point response below.

Referee #1:

This manuscript describes a potential role of NOD1/2 in sensing cellular stress via intracellular sphingosine-1-phosphate (S1P).

Although drugs employed effect different cell functions, the majority of the work was done focusing on the chemical compounds that induce ER stress or inhibition of protein synthesis. The concept that NOD1/2 are involved in sensing ER stress or protein misfolding is not novel.

Nevertheless, in previous reports both processes were driven by engagement with external bacterial peptidoglycan components rather than self-components.

This whole study is based on in vitro experiments performed with cell lines or in vitro established cultures of primary cells. Moreover, endogenous NOD1/2 expression appears not to be sufficient since the authors employed different strategies to increase their expression (e.g. HeLa cells were transfected with constructs for inducible NOD1 and NOD2 expression, BMDM were pretreated with IFN-g). This raises a major concern regarding to what degree the findings described have any real physiological relevance.

Author (AU): We thank the reviewer for acknowledging the novelty of our study, notably identification of an endogenous metabolite-S1P as an activator for NOD1/2. This new concept is a cornerstone for (1) linking cellular stress and sphingolipid metabolism to NOD1/2-mediated inflammation and for (2) uncovering roles of the S1P-NOD1/2 axis in pathological conditions where their cognate PAMP agonists appear obsolete, e.g. virus or parasite infections and chronic metabolic diseases. We are grateful for the constructive criticisms which we have addressed to our best.

Usage of several distinct cells (HeLa, human primary fibroblasts, THP-1, human CD14+ monocytes and BMDMs) enabled us to confidently conclude that S1P-NOD1/2 pathway is conserved in cells of different origins and in cells from different species. Moreover, we had to employ different cells according to requirements of specific experiments. For the lipidomics profiling, it is preferable to use primary fibroblasts, because (1) sphingosine metabolism is skewed in cancer cell lines (Ogretmen, 2018 PMID: 29147025; Ryland et al., 2011 PMID: 21209555); (2) human primary fibroblasts express both NOD1 and NOD2 (Hirao et al., 2009 PMID: 19734466; Jeon et al., 2012 PMID: 22218461; Uehara and Takada, 2007 PMID: 17314257). We also employed NOD1/2 double knockout (KO) and RIPK2 KO BMDMs from mice to confirm the requirement of NOD1/2 and RIPK2 in stress-induced inflammation. IFN-g priming was used to enhance the expression of RIPK2 in BMDMs, which is very low at basal

conditions (Kobayashi K et al. *Nature* 2002 PMID: 11894098, Kim Yun-Gi et al. *Immunity* 2008 PMID: 18261938, Dufner A et al. *Mol Cell Biol* 2008 PMID: 18160713). IFN-g priming has been widely used by others in this field (Nachbur U et al. *Nature Com.* 2015 PMID: 25778803, Keestra-Gounder AM et al. *Nature* 2016 PMID: 27007849). Therefore, we believe that the data obtained with various cells (overexpression vs knockout, mice vs human, cell lines vs primary cells) provide convincing evidence showing that the S1P-NOD1/2 axis is critical for stress-induced inflammation.

To further understand the physiological relevance of the S1P-NOD1/2 axis during infection, we performed infection experiments with *Shigella flexneri*. This pathogen triggers peptidoglycan-induced NOD1/2 activation, and induces actin cytoskeleton perturbation via its effectors and thereby activates NOD1. We observed that IL8 production induced by *Shigella flexneri* was impaired by SPHK1/2 dKD, indicating that S1P generation contributes to *Shigella*-induced inflammation. These results indirectly address the role of S1P in NOD1/2 activation upon bacterial infection. These novel data are shown in the Fig. 3J and Fig. S4J and described in the text lines 160-165.

Moreover, different aspects of this work were tested using different cell types. For example, cycloheximide is among the best inducers of IL-6 in NOD transfected HeLa cells, but MIP-2 production by BMDM is highest after tunicamycin exposure. Lipidomic profiling was performed on human primary fibroblasts, in which stimulation with cycloheximide indeed induced a lot of S1P but tunicamycin not so much. Indeed, the former is the best trigger of S1P and the latter less so.

While in this study sphk2 was required for secretion of inflammatory cytokines in response to cell stress, Sphk2 is known as a down-modulator of inflammatory responses since deficient myeloid cells display increased inflammatory cytokine production after LPS stimulation. The authors need to at least discuss this apparent discrepancy.

AU: We are grateful to the reviewer for raising these relevant issues which we have addressed by providing new experimental data and in-depth discussion. We also noticed the discrepancy between fibroblasts and BMDMs. We consider that this is due to the different sensitivity of these two cell types to cycloheximide (CHX) and tunicamycin. The same concentration of CHX induced a higher cell death in BMDMs than in fibroblasts. We performed additional experiments to quantify the abundance of cytosolic S1P in primary fibroblasts upon stimulation with various stressors. Consistent with the lipidomics analysis, various stresses increased the total abundance of S1P up to 3 μ M. CHX treatment resulted in the highest level of S1P in fibroblasts, correlating the highest IL6 production. These results are shown in the new Fig. 3C and are described in text at lines 143-144.

The anti-inflammatory role of SPHK2 during macrophage activation, namely roles in LPS tolerance (LPS stimulation for 24, 30 hours), is due to its interactions with TRAF6 and a block in recruitment of TRAF6 to mitochondria. This in turn inhibits mitochondrial ROS production and subsequently NF- κ B activation (Weigert A et al. *BBA-Molecular and Cell Biology of Lipids* 2019 PMID: 31128248). However, another study shows that SPHK1 and SPHK2 are not required for macrophage inflammatory responses *in vitro* and *in vivo* upon short stimulation of LPS (Xiong Y et al. *J Biol Chem.* 2013 PMID: 24081141). Moreover, several studies have demonstrated that SPHK2 is required for inflammatory response upon different stimulations (Ghosh M et al. *PLoS One* 2018 PMID: 29518138, Yang G et al. *Inflammation* 2018 PMID: 29728804). Thus, there is no consensus regarding SPHK2 as a general negative regulator of inflammatory responses and context dependent effects cannot be excluded. We provide evidence that S1P activates NOD1/2 and thereby support a role of SPHKs as positive regulators of inflammation. This is discussed in the text lines 294-300.

The characterization of S1P binding to NACHT/NBD domain of NOD1/2 that results in Rip2K activation is interesting but needs additional experiments. It is not clear if S1P binding interferes with NOD1/2 dimerization that is essential for Rip2K activation. Does it display ADP? Since ADP-mediated interaction between the central nucleotide-binding domain (NBD) and the winged-helix domain (WHD) is critical for stabilizing the closed conformation of NOD1/2, is it possible that S1P binding to NOD1/2 triggers or accelerates NOD degradation? Does S1P affect binding of the respective bacterial ligands to NOD1/2?

AU: We thank the reviewer for these insightful comments. In view of the available literature, we respectfully argue that NOD dimerization is still a matter of debate. However, the interaction with RIPK2 is well documented and represents a reliable and generally accepted readout for NOD1/2 activation. In our study we performed co-immunoprecipitation and showed that S1P induced recruitment of RIPK2 to NOD1 and NOD2 (Fig. 4C and Fig. 5A, 5B).

Following reviewer's suggestion, we addressed the interplay between ADP and S1P using MST assays. This technology allowed us to measure the binding of S1P to NOD1/2 in the presence or absence of ADP, and vice versa. These novel experiments demonstrated that S1P and ADP competed for interaction with NOD1/2 (please see the new Fig. 7A-7D, lines 265-272). In addition, we purified NOD1 H517A and performed MST assays to evaluate its binding affinity to S1P and ADP. The binding of both S1P and ADP to NOD1 was impaired by H517A mutation, further confirming that S1P and ADP interact with NOD1/2 at the same position (new Fig. S8I, Fig. 7E and 7F, lines 272-277). Thus, we propose that S1P displaces ADP from NOD1/2 and induces NOD1/2 activation possibly by promoting their active conformation. We discussed this in the text lines 339-347.

Following reviewer's suggestion, we investigated whether S1P destabilized NOD1/2 by performing additional experiments. We did not observe an obvious degradation of NOD1/2 upon S1P and iE-DAP/MDP stimulation. These data are presented as the new Fig. S6G and described in the text lines 182-183.

In silico docking and experiments with truncation mutants suggested that S1P interacted with the NBD domain of NOD1/2 (Fig. 6A and 6F). To further elucidate the different binding regions of S1P and iE-DAP/MDP, MST assays were performed (new Fig. 6G and 6H). The binding affinity of iE-DAP/MDP to NOD1/2 was not affected by S1P, demonstrating that iE-DAP/MDP and S1P use different binding regions of NOD1/2 (new Fig. 6G and 6H, lines 242-248). Co-stimulation with iE-DAP/MDP and S1P induced higher IL6 production than single stimulation alone, further confirming different activation mechanisms employed by S1P and iE-DAP/MDP (new Fig. S8E and S8F, line 248-253). Finally, we obtained new data showing that the Crohn's disease associated mutant NOD2 1007fs, which is unable to sense MDP and shows higher autoactivation, still can be activated by S1P. This data is presented as new Fig. S8G and describe in text lines 253-257.

While NOD1 is broadly expressed among hematopoietic and non-hematopoietic cells, expression of NOD2 is significantly more restricted. Thus, the concept that either has the same/equally role as a general cell stress-sensing function is somewhat surprising.

AU: NOD1 is thought to be widely expressed and NOD2 is also present in non-hematopoietic cells, for instance in fibroblasts (Hirao et al., 2009 PMID: 19734466; Jeon et al., 2012 PMID: 22218461; Uehara and Takada, 2007 PMID: 17314257). Hence, activation of NOD1 and NOD2 by S1P represents a general and broad pathway which links the cellular stress with inflammation.

We agree with the reviewer that the specific outcome of stress detection by these NLRs may be driven by their tissue expression and by the peculiar biology, including different signaling and effectors of these cells, e.g. non-hematopoietic vs. hematopoietic cells. As shown in Fig.

2 and Fig. 3, BMDMs mainly produce CXCL2 upon various stressors. Yet for fibroblasts, they are characterized with high IL6 production. This is discussed in the revised manuscript at lines 324-327.

Referee #2:

In the manuscript "The endogenous metabolite sphingosine-1-phosphate (S1P) activates NOD1/2-1 dependent inflammation triggered by cellular stress" by Pei et al, the authors report a role of NOD1/2 in surveillance of cellular stress through direct sensing of cytosolic metabolite S1P. Previous studies have demonstrated the involvement of S1P signalling in regulating heat shock proteins and the unfolded protein response in the ER and ER stress (i.e. Park et al, 2015). Furthermore, NOD1/2 roles in detecting the cellular stress have also been reported (i.e. Keestra-Gounder et al., 2016). In this regard, this new study could potentially add to the list of the very few known S1P molecular targets (i.e. TRAF2, cIAP) and also fill in a significant gap in the cell biology of NOD1/2-dependent cellular stress response. In order to strengthen the conclusions and increase the impact of this work, some additional controls and clarifications are required, however I believe the authors are well-positioned to address the comments.

AU: We thank the reviewer for the overall positive assessment and for recognizing the novelty of our work.

Major concerns

1) Most of the studies assessing SP1 induced NOD1/2 signaling involved overexpression systems in cell lines using digitonin at a final S1P concentration of 10 μ M. For a meaningful conclusion, the authors are encouraged to measure what is physiological S1P concentration that can be achieved under various cellular stressors such as tunicamycin or thapsigargin.

AU: We thank the reviewer for the excellent suggestion. We measured the levels of cytosolic S1P by ELISA. Stimulation of human primary fibroblasts with various stressors lead to the increase of cytosolic S1P abundances up to 3 μ M. These new data are presented in Fig. 3C and described in the text lines 143-144. Several reports have demonstrated that SPHK1/2 are targeted to ER or mitochondria upon different stresses, such as mitochondrial stress and serum starvation (Maceyaka M et al. J Biol Chem. PMID: 16118219, Kim S and Sieburth D Cell Rep. PMID: 30208318). Therefore, it is conceivable that the intracellular concentration of S1P at various subcellular sites is even higher than the overall levels that we measured. Thus, we consider that endogenous S1P is sufficient to induce NOD1/2 activation. In view of these new data, the concentration used for most of the experiments (10 μ M) likely reflects endogenous S1P levels. This has been discussed in the revised manuscript at lines 317-322.

2) It is now well established that NOD1/2 are peripheral membrane proteins and the endosomal membranes are the sites of peptidoglycan-induced NOD1/2-dependent signal transduction (Irving et al., 2014; Nakamura et al., 2014), which surprisingly the authors did not discuss. Particularly concerning is that their studies are done mostly using overexpressing cell lines without appropriate controls. For instance, the control would be a cell line overexpressing Crohn's disease associated NOD2 mutant (i.e. 3020insC; Nakamura et al., 2014) that is expected to be unable to respond to MDP stimulation while having an intact NACHT domain. This is also a concern because overexpression of the CARD(S) domain of NOD1 and NOD2 is sufficient to induce NF- κ B.

AU: We are grateful for these comments and suggestions which are critical for the understanding of NOD1/2 biology. We followed this suggestion discussed the membrane localization of NOD1/2 and its implication for our results (lines 362-367). NOD1 H517A mutant is supposed to mainly reside in the cytosol (Zurek B et al. Innate immune. PMID:

21310790). Although this mutant is not activated by direct stimulation with triDAP (Zurek B et al. *Innate immune*. PMID: 21310790), it is still activated upon cytosolic delivery of iE-DAP with digitonin (Fig. S8H, line 261-263). NOD2 1007fs is also present in the cytosol and defectively detects MDP (Barnich N et al. *J Cell Biol*. 2005 PMID: 15998797, Lecine P et al. *J Biol Chem*. 2007 PMID: 17355968). Cytosolic S1P still induced IL8 production in cells expressing NOD2 1007fs (new Fig. S8G). Therefore, cytosolic NOD1/2 are still responsive to their ligands. We have discussed our findings integrating relevance of NLR subcellular localization at lines 362-367.

We are well aware that expression of NOD1/2 and their CARD domains can induce auto activation. Indeed, we see autoactivation also in our stable cell lines. The cells expressing CARDS are not responsive to S1P stimulation. These results exclude the possibility that S1P induced NOD1/2-mediated NF-KB activation is due to self-activation (please see Fig. 6F). Moreover, we want to emphasize that we carefully characterized these cell lines and they still specifically respond to NOD1/2 agonists (please see Fig. S1B and S1C). In addition, we used distinct cells with endogenous expressed NOD1/2 and performed stimulation experiments (please see Fig. 4E-4K). The findings obtained with these cells further strengthen our results and rule out a singularly effect due to autoactivation.

According to reviewer's suggestion, we performed novel experiments with NOD2 1007fs as an additional control. S1P, other than MDP, induces IL8 production in cells expressing NOD2 1007fs, please see the new Fig. S8G, lines 253-257.

3) Seemingly, the predicted S1P binding site coincides with the nucleotide binding pocket (Fig 6A). Thus, the authors mention that H517 in the winged-helix (WH) domain is critical for S1P binding and function (Fig S8E). However, the same amino acid was shown to be critical for nucleotide binding and it was also shown to affect subcellular localization when mutated (Zurek at al, 2011). This needs clarification.

AU: We thank the reviewer for bringing up this important point. In the revised version of our manuscript, we address this by providing additional experimental data. MST experiments with NOD1 H517A showed that the binding of both S1P and ADP is compromised by the H517A mutation (please see the new Fig. S8I, Fig. 7E and 7F, lines 272-277). This is consistent with the data showing that S1P and ADP compete for the binding to NOD1/2 (please see the new Fig. 7A-7D, lines 265-272). The additional data sets strengthen our initial findings and are in line with the interpretation of the reviewer.

NOD1 H517A mainly resides in the cytosol. Cytosolic delivery of iE-DAP with digitonin induces the activation of NOD1 H517A, however, this mutant is not responsive to the stimulation of cytosolic S1P (please see Fig. S8H, lines 261-263)

Therefore, we conclude that S1P binding to H517 replaces ADP from NOD1/2 and thus induces NOD1/2 activation by promoting their active conformation. This has been discussed in the revised manuscript at lines 339-347.

4) In addition, it was shown that the NOD1 affinity for ATP and ADP is in nanomolar range (IC50 of ~15 nM for ATP and ~190nM for ADP) (Askari et al, 2012). The authors MST results showed that the NOD1/2 affinity for S1P is low micromolar range (Fig 6C,D and E). Competitive displacement experiments to determine if the S1P binding is affected by ATP or ADP binding are required to support the conclusions.

AU: Following reviewer's excellent suggestion, we performed MST assays to examine the binding of S1P to NOD1/2 in the presence or absence of ADP, and vice versa. The results demonstrate that S1P and ADP compete for the binding to NOD1/2. Please see the new Fig. 7A-7D, line 265-272.

5) Despite the analysis performed in this study, the question remains as to whether S1P binding to NOD1/2 is specifically responsible for S1P-induced NOD1/2-dependent signaling and it's not an indirect effect (see point #1). If the authors are correct, then the authors should be able to generate a mutant NOD1/2 protein that does not bind to SP1 but it's still able to bind ATP. This mutant should be introduced into NOD-deficient cells and examined for S1P induced signaling and localization.

AU: Based on the MST data that S1P and ADP compete for binding to NOD1/2 (new Fig. 7A-7D, line 265-272), we conclude that S1P and ADP bind to the same region of NOD1/2. This was further confirmed by MST assays showing that H517A mutant is defective on binding to both S1P and ADP (new Fig. S8I, Fig. 7E and 7F, lines 272-277). Therefore, we assume that it is not feasible to generate a mutant that does not bind to S1P, but maintain its interaction with ADP/ATP.

6) The experiments to show interaction between NOD1/2 and SP1 (Fig 5A,B) should be improved. The analysis should include analysis of wild-type and predicted point mutations of SP1 binding site with or without ATP/ADP.

AU: We respect the view of the reviewer, however we would like to emphasize that we used several independent assays (dSTORM, single particle tracking, MST assays, co-IP) to assess the interaction of NOD1/2 with S1P. Further, in the revised version of the manuscript we included additional MST experiments with H517A and demonstrated that the binding of S1P and ADP is impaired by the H517A mutation (new Fig. S8I, Fig. 7E and 7F, lines 272-277).

Minor points

1) Did the authors observe any synergistic effect of S1P and iE-DAP/MDP co-stimulation on NOD1/2 dependent signaling?

AU: We thank the reviewer for the excellent suggestion. Co-stimulation of iE-DAP/MDP with S1P induced higher IL6 production than single stimulation alone, suggesting additive effects of S1P and iE-DAP/MDP. These experimental data is shown in the new Fig. S8E and S8F and described in the text at lines 248-253.

2) Fig S1A. MWs of NOD1 and NOD2 appears to be the same.

AU: We have run gels with better separation and changed the gel image. Please see the new Fig. S1A.

3) I could not find details about the lipid coated beads.

AU: We apologize for this omission. S1P-coated beads (S-6110-2) and other sphingolipids coated beads (P-B00SS) were obtained from Echelon Biosciences. This is included in the Material and Methods section of the revised manuscript. Please see the revised manuscript, lines 886-887.

4) The authors should include more detail about their molecular dynamic analysis.

AU: We thank the reviewer for this suggestion. Additional details are provided in the Material and Methods section of the revised manuscript. Please see text lines 1111-1132.

Referee #3:

Pei and colleagues identified the cytosolic immune receptors for peptidoglycan fragments as novel sensors for various unrelated stressors. In this regard, Nod-activation required

endogenous cytosolic sphingosine-1-phosphate production. The authors present extensive experimental work using multifarious methods.

Major concerns:

- Order and number of the used reagents differ between experiments.

AU: We thank the reviewer for carefully reading the manuscript and for the constructive criticisms. In the revised version of the manuscript, we unified the order of number of the used reagents in Fig. 1 and Fig. S1 (please see the new Fig. S1B-S1G). Based on the modes of action, we divided these stressors into different classes: microtubule/actin network perturbation, Golgi stress, protein synthesis block, mitochondrial stress, ER stress and DNA damage. To simplify, we only chose 1 or 2 reagents from each class in the rest of experiments.

- All experiments are performed under basal conditions. However, NOD-receptors have their specific immune function especially during infection. Are the identified mechanisms also relevant under these conditions? This should be investigated, at least exemplarily.

AU: We are grateful for these comments and welcomed the suggestion to address exemplary the relevance of the S1P/NOD pathways during infection. We performed Shigella infection experiments with HeLa SPHK1/2 dKD cells. IL8 production induced by Shigella flexneri was impaired by SPHK1/2 dKD, suggesting that S1P generation is required for Shigella induced inflammation. These results suggest the role of S1P in NOD1/2 activation upon bacterial infection. These novel data are shown in the Fig. 3J and Fig. S4J and described in the revised manuscript at lines 160-165.

- Figure 1A,B: More or less all used stressors were effective to inhibit NOD-dependent IL6... are there stressors being not effective? Such controls are needed. Maybe TLR-agonists. Are the detected differences statistically different? As well in many other experiments, e.g. Fig. S2.

AU: In Fig. 1A, B, doxycycline was used to induce the expression of NOD1/2. It was effective to induce NOD1/2 expression as shown in Fig. S1A. They still specifically respond to NOD1/2 agonists (please see Fig. S1B and S1C). Selected stressors effectively induced IL6 production in a NOD1/2 dependent manner. As a control, both BFA and monensin (inhibitors of ER-Golgi trafficking and protein secretion) did not induce IL6 production (Fig. 1A-1D). This suggests that various stressors specifically induce IL6 production via NOD1/2 activation.

We have conducted statistical analysis for Fig. 1A and Fig. 1B according to the reviewer's suggestion. P values were calculated with ANOVA test of linear mixed model for fix effect of main factors (Lines 639-640). We also performed statistical analysis for Fig. S2. Serum depletion did not block IL6 production upon most of stimulations, excluding the possibility of serum contamination with peptidoglycan. For inhibition of endocytosis with dynasore, most of the stimulations were not affected. However, paclitaxel, nocodazole, cytochalasin D and actinonin even induce higher IL6 production in the presence of dynasore. As a control, iE-DAP/MDP stimulation without digitonin was impaired by dynasore, demonstrating the effectiveness of dynasore (please see the new Fig. S2B-S2E).

- Fig. 1: Compared to Fig. S1 order of columns is different, this is confusing.

AU: The order of Fig. S1 has been changed accordingly. Thank you for this suggestion.

- How often were the Western blots performed? A quantification of the blots would be helpful.

AU: We apologize for not giving this information during the initial submission. Each immunoblot is representative of at least three independent experiments. Quantification of the blots was added in the revised version of the manuscript, please see the new Fig. S3D, Fig. S4I and Fig. S7G. Thank you for this suggestion.

Minor concerns:

- Results page 11, below: Fig. S9B, S9C should be Fig. S8B, S8C.

AU: We apologize for this mistake, which was corrected in the revised version.

Dear Stefan,

Thank you for submitting your revised manuscript to The EMBO Journal. Your study has now been seen by the three referees and their comments are provided below. Referee #1 is not fully satisfied that the physiological significance aspect has been sorted out. However, I do appreciate the introduced changes and find the manuscript important and insightful.

I would therefore like to ask you to address the last few specific comments in a final revision. I don't think any further experiments are needed just better clarifications.

When you submit your revised version will you also take care of the following points:

We need 3-5 keywords

Please check the reference list - there are some citations where there is more than 10 authors listed

Can you double check that all the funding/grants info is also listed in the MS

The figure files need to be uploaded as individual figure files.

Please check the order of the manuscript sections.

You have at the moment 9 supplemental figures they should be either turned into EV figures or added to an appendix. Note we can only have 5 EV figures. Please also make sure to correct figure callout. See author guidelines

<https://www.embopress.org/page/journal/14602075/authorguide#expandedviewwhich>

We encourage the publication of source data, particularly for electrophoretic gels and blots, with the aim of making primary data more accessible and transparent to the reader. It would be great if you could provide me with a PDF file per figure that contains the original, uncropped and unprocessed scans of all or key gels used in the figure? The PDF files should be labeled with the appropriate figure/panel number, and should have molecular weight markers; further annotation could be useful but is not essential. The PDF files will be published online with the article as supplementary "Source Data" files.

We include a synopsis of the paper (see <http://emboj.embopress.org/>). Please provide me with a general summary statement and 3-5 bullet points that capture the key findings of the paper.

We also need a summary figure for the synopsis. The size should be 550 wide by [200-400] high (pixels). You can also use something from the figures if that is easier.

I have asked our publisher to do their checks on the paper. They will send me the file within the next few days. Please wait to upload the revised version until you have received their comments.

Please upload a point-by-point response also concerning the editorial points above

Let me know if we need to discuss anything further.

With best wishes

Karin

Karin Dumstrei, PhD
Senior Editor
The EMBO Journal

Further information is available in our Guide For Authors:

The revision must be submitted online within 90 days; please click on the link below to submit the revision online before 10th Jun 2021.

Referee #1:

The authors improved some aspects of the manuscript (e.g. by providing better characterization of the S1P with NOD1/2 interaction). Nevertheless, the study remains exclusively based on in vitro experiments (e.g. the new "infection" experiment consists of infecting HeLa cells with Shigella).

As the authors state in the rebuttal letter, "we had to employ different cells according to requirements of specific experiments" and, when there are differences in response among cell lines, they argued that this was "due to different sensitivity" of cell lines. Thus, my main reservation regarding whether the findings have any real physiological relevance remains.

Fig. 3J lacks statistical analysis.

The data from 3 different experiments are shown and it appears that there is significant variability among the results from independent tests. It would be better to show the findings of each assay separately.

Fig. S8 E-F, x axis indicates that NOD1/2 ligands were only (but twice) tested in the presence of S1P.

Regarding the experiment shown in S8 E-F: Since S1P competes with the ATP-binding site of NOD1/2, it could act as an antagonist of the signal (inflammation) triggered by NOD1/2 ligands. The synergistic effect may be observed only in a transfected cell system where the amount of available NOD1 or NOD2 is not a limiting factor.

In Fig. S8 G, 60 % of the response is due just to just HEK293 transfection.

While in Fig. S8 E-F HeLa transfected cells were stimulated for 20 hours, in Fig. S8 G supernatants from HEK293T transfectants were tested after 48 hours of stimulation.

Referee #2:

The new results are convincing. The authors are to be commended for their job of responding to the critiques, which has markedly improved this story.

Referee #3:

For the most part, the authors have adequately addressed the comments. However, some issues should be revised prior publication.

- The authors clearly explained in the response to reviewers why different numbers of inhibitors were used throughout the figures ('To simplify, we only chose 1 or 2 reagents from each class in the rest'). This should be likewise explained in the manuscript text.
- Dots showing single data points are sometimes too big and partly cover the column behind, e.g. Fig. 2D, better e.g. in Fig. 3E.
- Fig. 3C: What is meant by $p=0.054$? Which condition?
- Fig. 3J: $n=2$? This is not an academic standard.
- Why are single data points sometimes shown, sometimes not...? ELISA yes, PCR not, but not strictly, see e.g. Fig. 4F/G and Fig. 6F. This should be consistent.

- Please check again the order of the used inhibitors because it is still not consistent, e.g. Fig. 3G/H/I.

Ref: EMBOJ-2020-106272

Title: The endogenous metabolite sphingosine-1-phosphate (S1P) activates NOD1/2-dependent inflammation triggered by cellular stress

We thank the editor and reviewers for their time, thoughtful comments, and appreciate the opportunity to respond. In our revision, we have included key words, synopsis, expanded figures, source data and author checklist, revised the order of the main text, figure legends and the references according to the publisher's suggestions. According to the reviewers' constructive comments, we have repeated the *Shigella* infection experiment and presented the data as revised Figure 3J. We also revised other issues according to reviewers' comments. Please find our point-by-point response below.

Referee #1:

The authors improved some aspects of the manuscript (e.g. by providing better characterization of the S1P with NOD1/2 interaction). Nevertheless, the study remains exclusively based on *in vitro* experiments (e.g. the new "infection" experiment consists of infecting HeLa cells with *Shigella*).

As the authors state in the rebuttal letter, "we had to employ different cells according to requirements of specific experiments" and, when there are differences in response among cell lines, they argued that this was "due to different sensitivity" of cell lines. Thus, my main reservation regarding whether the findings have any real physiological relevance remains.

Author (AU): We thank the reviewer for acknowledging the improvement of our manuscript. Concerning the different cells used in our study, the majorities of all the stimuli (cytochalasin D, nocodazole, vinblastine, cytoheximide, etoposide, CCCP, etc) display similar effects across primary cells or cell lines/different cells types/various species, demonstrating that the S1P-NOD1/2 axis is universally critical for stress-induced inflammation.

The *in vitro* infection experiment of HeLa cells with *Shigella* brings physiological relevance as it demonstrates the necessity of the newly identified pathway (S1P-NOD1/2) for inflammatory responses triggered by the pathogen. Unfortunately, there is no reliable mouse model for *Shigella* / shigellosis. Moreover, single knockout of *Sphk1* or *Sphk2* results in no significant reduction or even elevation of S1P levels (PMID: 15459201, PMID: 29029455). Further, *Sphk1/2* double knockout causes embryonic lethality (PMID: 16314531). Thus, we are restricted to *in vitro* studies and hope that the reviewer understands the current limitations. We intend to extend our findings to additional models and this is the scope of future studies.

Fig. 3J lacks statistical analysis. The data from 3 different experiment are shown and it appears that there is significant variability among the results from independent tests. It would be better to show the findings of each assay separately.

Author (AU): We thank the reviewer for raising this issue. We have repeated this experiment to include statistical analysis. The combined data is shown in the revised Figure 3J and substantiates our initial conclusions.

Fig. S8 E-F, x axis indicates that NOD1/2 ligands were only (but twice) tested in the presence of S1P. Regarding the experiment shown in S8 E-F: Since S1P competes with the ATP-binding site of NOD1/2, it could act as an antagonist of the signal (inflammation) triggered by NOD1/2 ligands. The synergistic effect may be observed only in a transfected cell system where the amount of available NOD1 or NOD2 is not a limiting factor.

AU: We apologize for the confusion caused by the mislabeling of x axis. We amended the labels of the x-axis accordingly. 3 independent experiments were performed. Co-stimulation of S1P together with different concentration of iE-DAP/MDP (3, 20, 50 μ M) induced higher IL6 production compared to single stimulation alone, even at low concentration (3 μ M). This does not support an antagonist / inhibitory role of S1P.

In Fig. S8 G, 60 % of the response is due just to just HEK293 transfection.

AU: We thank the reviewer for this insightful comment. Transfection of NOD1/2 in HEK293T cells already induces IL8 production and in particular the 1007fs induces high autoactivation (PMID: 15044951). In any case our cells expressing NOD2 1007fs are still responsive to S1P stimulation.

While in Fig. S8 E-F HeLa transfected cells were stimulated for 20 hours, in Fig. S8 G supernatants from HEK293T transfectants were tested after 48 hours of stimulation.

AU: We thank the reviewer for bringing up this issue. HeLa cells produce large amounts of IL6 at 16-20 hours post stimulation. However, compared to HeLa cells, the abundance of IL8 in HEK293T cells is much lower and therefore the incubation time was extended.

Referee #2:

The new results are convincing. The authors are to be commended for their job of responding to the critiques, which has markedly improved this story.

AU: We are grateful for the comments and suggestions that improved our manuscript and we thank the reviewer for the positive assessment.

Referee #3:

For the most part, the authors have adequately addressed the comments. However, some issues should be revised prior publication.

- The authors clearly explained in the response to reviewers why different numbers of inhibitors were used throughout the figures ('To simplify, we only chose 1 or 2 reagents from each class in the rest'). This should be likewise explained in the manuscript text.

AU: We thank the reviewer for this suggestion. We revised the manuscript accordingly and introduced this explanation in the main text lines 109-111.

- Dots showing single data points are sometimes too big and partly cover the column behind, e.g. Fig. 2D, better e.g. in Fig. 3E.

AU: We thank the reviewer for this comment. We revised the figures (Fig. 1C, Fig. 1D, Fig. 2D, Fig. 3E-3G, Fig. 3I, Fig. 4G-4K) according to the reviewer's suggestion to keep the sizes of the data points similar and improve comprehensibility of the figures.

- Fig. 3C: What is meant by $p=0.054$? Which condition?

AU: The p value (control vs tunicamycin) was calculated using one-way ANOVA with Holm-Šidák's multiple comparisons test.

- Fig. 3J: n=2? This is not an academic standard.

AU: We repeated the infection experiment and included statistical analysis. The combined data is shown in the revised Figure 3J and substantiates our initial conclusions.

- Why are single data points sometimes shown, sometimes not...? ELISA yes, PCR not, but not strictly, see e.g. Fig. 4F/G and Fig. 6F. This should be consistent.

AU: We thank the reviewer for this comment. The decision to include single data points in specific assays is based on visualization of the data. For the qPCR measurements, we decided against individual data points because the graphs would be very difficult to read with many groups. However, we included data points for all of the ELISA experiments.

- Please check again the order of the used inhibitors because it is still not consistent, e.g. Fig. 3G/H/I.

AU: We thank the reviewer from carefully checking for consistency and apologize for being inconsistent. We revised Fig. 3G/3H/3I to keep the order of the inhibitors consistent.

Dear Stefan,

Thank you for submitting your revised manuscript to The EMBO Journal. I have now had a chance to take a look at it and I appreciate the introduced changes.

I am therefore very pleased to accept the manuscript for publication here.

Congratulations on a nice study!

with best wishes

Karin

Karin Dumstrei, PhD
Senior Editor
The EMBO Journal

Please note that it is EMBO Journal policy for the transcript of the editorial process (containing referee reports and your response letter) to be published as an online supplement to each paper. If you do NOT want this, you will need to inform the Editorial Office via email immediately. More information is available here: https://emboj.embopress.org/about#Transparent_Process

Your manuscript will be processed for publication in the journal by EMBO Press. Manuscripts in the PDF and electronic editions of The EMBO Journal will be copy edited, and you will be provided with page proofs prior to publication. Please note that supplementary information is not included in the proofs.

Should you be planning a Press Release on your article, please get in contact with embojournal@wiley.com as early as possible, in order to coordinate publication and release dates.

If you have any questions, please do not hesitate to call or email the Editorial Office. Thank you for your contribution to The EMBO Journal.

Corresponding Author Name: Stefan H.E. Kaufmann, Anca Dorhoi and Gang Pei

Journal Submitted to: EMBO J

Manuscript Number: EMBOJ-2020-106272